# Photic zone niche partitioning, stratification, and carbon cycling in the tropical Indian Ocean during the Piacenzian

Deborah N. Tangunan[1,2], Ian R. Hall[1], Luc Beaufort[3], Melissa A. Berke[4], Alexandra Nederbragt[1], Paul R. Bown[2]

[1]School of Earth and Environmental Sciences, Cardiff University, Cardiff, United Kingdom

[2]Department of Earth Sciences, University College London, London, United Kingdom

[3]CNRS-CEREGE, BP 80, 13545 Aix-en-Provence Cedex 04, France

[4]Department of Civil and Environmental Engineering and Earth Sciences, University of Notre Dame, Notre Dame, IN, USA

*Correspondence to:* Deborah N. Tangunan (d.tangunan@ucl.ac.uk)

**Abstract.** The mid-Piacenzian Warm Period (mPWP; ~3.264–3.025 Ma) marks the most recent episode of sustained global warmth, characterised by atmospheric carbon dioxide ($p$CO$_2$) levels similar to those of today. Despite this, our understanding of the vertical structure of the Pliocene ocean and its role in modulating global carbon cycling during this period remains poorly resolved. Here, we combine planktic (coccolith and planktic foraminifera) and benthic (benthic foraminifera) stable carbon ($\delta^{13}$C) and oxygen ($\delta^{18}$O) isotope records from the International Ocean Discovery Program Site U1476 in the western tropical Indian Ocean (Mozambique Channel), to reconstruct surface-to-deep ocean conditions during the mPWP. The consistently high vertical gradients in $\delta^{13}$C and $\delta^{18}$O indicate long-term thermal stratification and increased carbon export in this moderately elevated $p$CO$_2$ world. Distinct isotopic signatures observed between the deep-photic zone coccolithophore *Florisphaera profunda* which dominates the coccolith assemblages, and mid-photic zone planktic foraminifera *Globigerinoides ruber* suggest ecological partitioning and differing sensitivities to upper ocean dynamics (e.g., stratification, nutrient supply, light intensity). A transient breakdown in stratification and deep ocean carbon storage during Marine Isotope Stage M2 (~3.30–3.28 Ma), a glacial interval preceding the peak warmth of the mPWP, demonstrates the vulnerability of the tropical ocean structure to high-latitude climate forcing. Spectral analysis reveals pronounced obliquity-paced variations in both $\delta^{13}$C and $\delta^{18}$O records, linking high-latitude orbital forcing to carbon cycling in low-latitude regions. These findings offer important new constraints on the ocean–atmosphere carbon feedback during the mPWP and underscore the previously underappreciated role of the tropical Indian Ocean as a dynamic component of global carbon cycling during past warm periods, providing a key low-latitude counterpart to better-studied Atlantic and Pacific regions.

## 1 Introduction

The mid-Piacenzian Warm Period (mPWP; 3.264–3.025 million years ago, Ma) represents a pre-Quaternary interval of global warmth, characterised by polar amplification, elevated sea surface temperatures (SSTs), and reduced ice volume (Dowsett et al., 2016). Sustained global warmth characterises the mPWP, contrasting with the progressively cooler, more variable climate of the Pleistocene, marked by intensified Northern Hemisphere glaciation and glacial–interglacial cycles. The mPWP occurred against a backdrop of long-term climate cooling and atmospheric carbon dioxide ($p$CO$_2$) concentrations of ~380 to 420 ppmv, comparable to modern levels (Bartoli et al., 2011; Seki et al., 2010; de la Vega et al., 2020), yet global mean temperatures were ~3 °C warmer than the pre-Industrial era (Salzmann et al., 2011; IPCC, 2023; Tierney et al., 2020). Unlike extreme greenhouse episodes in Earth's history, such as the Paleocene–Eocene Thermal Maximum (PETM; ~56 Ma), which involved abrupt warming of ~5–8 °C under $p$CO$_2$ levels likely exceeding 1000 ppm (Zachos et al., 2008), the mPWP represents a more moderate warming scenario (1.2–5.2 °C; Haywood et al., 2000). This makes it a uniquely relevant analogue for understanding climate system sensitivity under $p$CO$_2$ levels near modern concentrations.

The increase in global temperatures during the mPWP were the result of elevated atmospheric $p$CO$_2$ levels (de la Vega et al., 2020), a reduction in planetary albedo associated with sea ice loss and vegetation changes leading to increased absorption of solar radiation (Chandan and Peltier, 2018), and enhanced ocean heat transport (Raymo et al., 1996). Climate models from the Pliocene Model Intercomparison Project Phase 2 (PlioMIP2) indicate a stronger Atlantic Meridional Overturning Circulation (AMOC) during this interval (Weiffenbach et al., 2023), consistent with earlier findings of an intensified meridional overturning circulation in the North Atlantic (Raymo et al., 1996). More pronounced warming occurred at high latitudes due to polar amplification, driven by reductions in sea ice and snow cover, which lowered albedo and enhanced regional warming (Howell et al., 2014). Additionally, land surface and vegetation changes, particularly forest expansion and desert retreat, altered surface albedo and further contributed to global warming through feedback mechanisms (Zhang and Jiang, 2014). This warming led to rising sea levels due to reduced continental ice volume and thermal expansion of seawater (De Boer et al., 2014), significant alterations in both oceanographic and atmospheric circulation patterns, including intensified poleward heat transport and modified wind systems  (McClymont et al., 2023; McClymont et al., 2020), and triggered significant shifts in marine biodiversity and ecosystem functioning through changes in species distributions during the mPWP (e.g., Larina et al., 2025).

In light of ongoing concerns regarding anthropogenic CO$_2$ forcing, the mPWP has been identified as the most relevant (and nearest) past analogue for future warm climate (Haywood et al., 2020; Dowsett et al., 2013), making it an ideal interval for high-resolution studies of sustained warm conditions and the abrupt cooling events that interrupted the warm background state (e.g., marine isotope stages (MIS) M2, KM2, G20) (Dolan et al., 2011). Despite its relevance, a key controversy persists over the carbon cycling mechanisms that sustained this prolonged warmth under $p$CO$_2$ levels that, while elevated compared to Pre-Industrial values, remained lower than those associated with other major geological warm periods. Specifically, the

extent to which changes in ocean circulation and regional productivity contributed to global carbon storage and feedback mechanisms remains unresolved. Here we address this controversy by focusing on high-resolution records from the western tropical Indian Ocean. This region has been understudied, yet it was dynamically important during the Pliocene. The

International Ocean Discovery Program (IODP) Site U1476 (15° 49.25′ S; 41° 46.12′ E; 2166 m water depth) in the Mozambique Channel recovered a continuous carbonate-rich succession spanning the Plio-Pleistocene (Hall et al., 2017), providing unparalleled resolution for the Piacenzian stage.

Site U1476 is located on the Davie Ridge at the northern entrance to the Mozambique Channel (**Fig. 1a**). This region is

dynamically sensitive to both Indian Ocean thermocline dynamics and high-latitude water mass fluxes (Hall et al., 2017; **Fig. 1b**). The Mozambique Channel is bordered by Madagascar and the southeastern African margin, where surface and subsurface currents vary significantly due to vigorous eddy activity and interactions with the broader Indian Ocean circulation systems (De Ruijter et al., 2002; Schouten et al., 2003). This dynamic circulation feeds into the Agulhas Current, a major western boundary current that transports warm, saline waters southwestward toward the South Atlantic (Beal et al.,

2011), playing an essential role in regulating the AMOC, a key mechanism for global ocean heat and salt redistribution (De Ruijter et al., 2002; Schott et al., 2009). Presently, the western tropical Indian Ocean exhibits a near equilibrium between surface ocean and $p$CO$_2$, resulting in low net air–sea CO$_2$ exchange (Takahashi et al., 2009; Guo and Timmermans, 2024), underscoring the importance of Site U1476 in both ocean circulation and climate-carbon cycle feedback mechanisms.

Here we present high resolution (~ 2.25 kyr) stable carbon ($\delta^{13}$C) and oxygen ($\delta^{18}$O) isotope records from fine fraction bulk sediment (<20 μm, herein referred to as coccolith fraction) and benthic foraminifera, alongside lower resolution (average of ~25 kyr) $\delta^{13}$C and $\delta^{18}$O records from the planktic foraminifera *Globigerinoides ruber*, capturing significant changes in surface productivity, thermohaline restructuring, and deep-sea carbon accumulation (Zachos et al., 2001; Sigman and Boyle, 2000). We also present a high-resolution datasets of coccolith assemblages and morphometry from the same samples.

Together, these records offer a new opportunity to examine the oceanographic and biogeochemical feedback mechanisms that governed carbon cycling and sustained warmth during the mPWP.

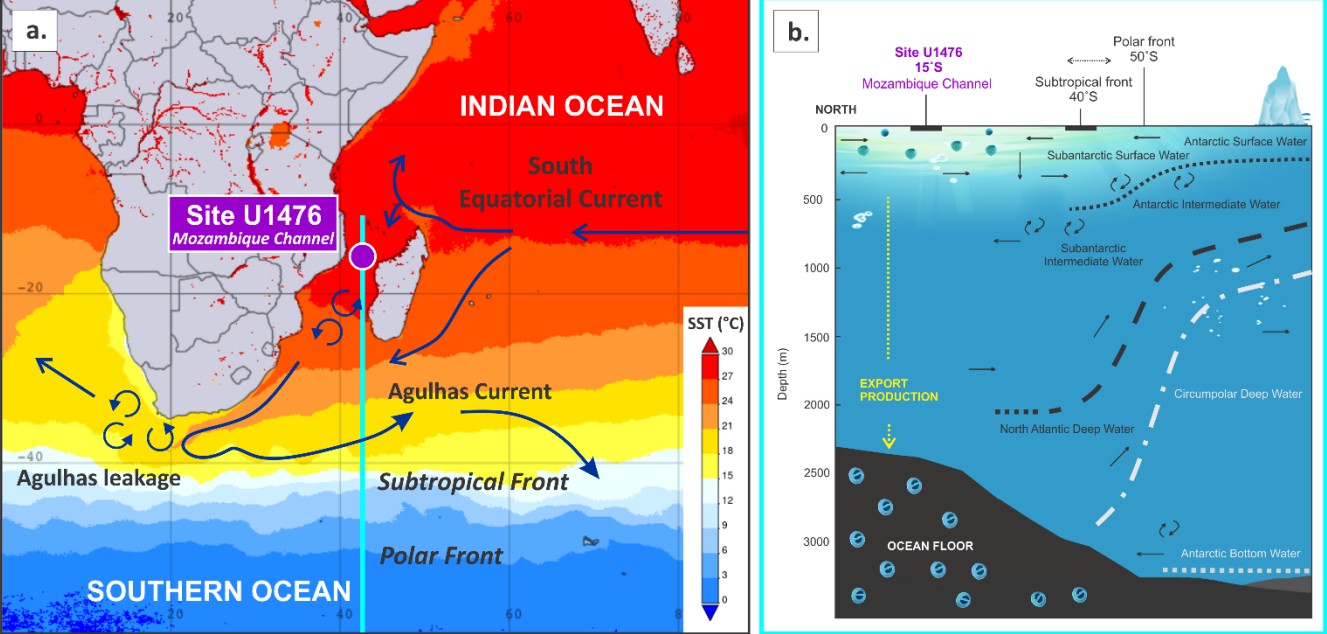

**Figure 1: (a)** Sea surface temperature (SST, °C; Acker & Leptoukh, 2007) and major currents in the Indian Ocean (Beal et al.,
2011), showing the location of IODP Site U1476 in the Mozambique Channel. **(b)** Schematic cross-section showing the position of
Site U1476 relative to major water masses (adapted from Westall and Fenner, 1991) and the Southern Ocean fronts.

## 2 Methods

### 2.1 Sampling strategy

Samples were selected between 79.80 m and 96.85 m depth on the stratigraphic splice, targeting the Piacenzian. Using the
shipboard chronology based on combined calcareous nannofossil, planktic foraminifera, and magnetostratigraphy (Hall et
al., 2017), samples were taken every 8 cm, yielding 218 samples and providing an average time resolution of ~2.25 thousand
years (kyr). $\delta^{18}O$ and $\delta^{13}C$ stable isotopes were analysed from all 218 samples for the coccolith fraction (<20 µm) and
benthic foraminifera, with 20 samples analysed for planktic foraminifera.

### 2.2 Age model

The age model for Site U1476 is based on tuning the benthic foraminifera $\delta^{18}O$ to the LR04 global stack (Lisiecki and
Raymo, 2005) using QAnalyseries (Kotov and Pälike, 2018) **(Fig. S1a–1b)**. The calibrated age model yielded sedimentation
rates that range between ~2.5 and 4.7 cm kyr$^{-1}$ (average ~3.5 cm kyr$^{-1}$) **(Fig. S1g–1h)**, consistent with the nearby Deep Sea

Drilling Project (DSDP) Site 242 (~2.5 cm kyr$^{-1}$; Simpson and Schlich, 1974), and with the shipboard estimate of 2.3 to 3.5 cm kyr$^{-1}$ (Hall et al., 2017).

## 2.3 Benthic foraminifera carbon and oxygen stable isotopes

Refrigerated samples were soaked in deionised (DI) water, placed in a spinner for 24 hours to disaggregate, and then wet-
sieved through a 60 μm mesh. The fine (<60 μm) and coarse (>60 μm) fractions were oven-dried at 40 ˚C. After drying, the >60 μm fraction was dry-sieved; three specimens of benthic foraminifera *Cibicidoides wuellerstorfi* (syn. *Lobatula wuellerstorfi*) were picked from the 250-355 μm size range. Individual foraminiferal tests were hand-picked under a binocular microscope, selecting the best-preserved and cleanest specimens. In intervals where there is not enough *C. wuellerstorfi*, other *Cibicidoides* species (i.e., *Cibicidoides bradyii*, *Cibicidoides mundulus*) or *Uvigerina* species were
chosen. The picked foraminiferal tests were gently rinsed in DI water and quickly dried. This gentle cleaning step aimed to dislodge any contaminating carbonate material (e.g., small/juvenile foraminifera, coccoliths, or detrital carbonate) adhering to test surfaces or trapped within apertures. Following rinsing and drying, the tests were crushed between two clean glass plates and homogenised for analysis.

$\delta^{18}O$ and $\delta^{13}C$ were measured using a Finnigan MAT 253 with a Carbonate Kiel IV autosampler, where each sample was reacted with 70 ˚C phosphoric acid ($H_3PO_4$). Analyses were performed at the Stable Isotope Facility, School of Earth and Environmental Sciences, Cardiff University (United Kingdom). Values are reported relative to the Vienna Pee Dee Belemnite (VPDB) scale and calibrated against NBS 19 and internal laboratory standards. The mean external reproducibility of carbonate standards is ±0.02 ‰ for $\delta^{13}C$ and ±0.03 ‰ for $\delta^{18}O$. For benthic foraminifera, only *C. wuellerstorfi* was used for
$\delta^{13}C$ analyses, while both *Cibicidoides* spp. and *Uvigerina* spp. were used for $\delta^{18}O$. $\delta^{18}O$ values from *Cibicidoides* spp. were adjusted by +0.64 ‰ to align with *Uvigerina* (Shackleton and Opdyke, 1973). $\delta^{13}C$ values from *Uvigerina* were excluded due to a well-known metabolic vital effect that causes them to record values approximately 1.0‰ lower than the ambient dissolved inorganic carbon (DIC; e.g., Zahn et al., 1986). When replicate analyses were possible, values are reported as the average of the results. The two highly negative $\delta^{13}C$ values (−0.83 and −0.62 ‰) were excluded from interpretation and
considered outliers, likely reflecting analytical or sample-related artefacts rather than original environmental signals.

## 2.4 Planktic foraminifera carbon and oxygen stable isotopes

From the same dry-sieved 250-355 μm fraction, we picked three specimens of planktic foraminifera *G. ruber*. *G. ruber* calcifies in the mixed layer between 20 and 50 m (Mohtadi et al., 2011) and represents the surface water column signal. $\delta^{18}O$
and $\delta^{13}C$ were measured using the same protocol as the benthic foraminifera, at the same facility.

## 2.5 Coccolith carbon and oxygen stable isotopes

A small amount of sediment (~1 g) was wet-sieved using a 20 μm mesh with buffered water (0.05 % ammonia diluted in DI water). The filtrate was collected and centrifuged to concentrate and obtain the desired fraction. The <20 μm fraction was placed in a watch glass and immediately dried on a hotplate at 50 ˚C, with acetone added to hasten evaporation. The dried material, which is considered to be all biogenic carbonate, was homogenised and then analysed for $\delta^{18}O$ and $\delta^{13}C$ following the same procedure described above for planktic and benthic foraminifera at the same facility. Smear slides of selected samples were prepared and checked under a light microscope to make sure that the material is composed of at least 95 % coccoliths. Coccolithophores calcify in the photic layer and represent the upper 200 m of the water column (Winter et al., 1994). We assume similar calcification patterns across all the species, although we did not disregard potential "vital effects" in interpreting the isotopic results (Stoll, 2005). Since coccolithophores are the primary carbonate producers in the investigated region, we consider the isotope record to provide a reliable picture of environmental variations in the upper ocean.

## 2.6 Calculation of isotopic gradients

To evaluate the vertical structure of the ocean, we calculated the isotopic gradients (Δ) between microfossil groups representing distinct depth habitats within the water column (**Fig. 2a**). Specifically, we measured the differences in $\delta^{13}C$ and $\delta^{18}O$ between planktic foraminifera and coccoliths, benthic foraminifera and coccoliths, and benthic and planktic foraminifera. These gradients serve as a first-order approximation of vertical isotopic offsets across the photic zone and between the surface and deep-ocean layers. While $\delta^{13}C$ and $\delta^{18}O$ gradients between benthic and planktic foraminifera are well-established indicators of ocean stratification and carbon cycling (e.g., Hodell and Venz-Curtis, 2006), the use of gradients involving coccoliths is less common. Nonetheless, studies by Hermoso et al. (2016; 2020) have shown that $\delta^{13}C$ offsets between coccoliths and planktic foraminifera are sensitive to physiological processes, such as vital effects linked to ambient $p\mathrm{CO_2}$ levels. Based on this framework, we interpret the isotopic gradients involving coccoliths, both with planktic and benthic foraminifera, as integrated signals of upper ocean carbon cycling, stratification, and potential vertical habitat partitioning driven by environmental variability.

## 2.7 Coccolith assemblage composition and abundance

Slides were prepared using the random settling technique (Beaufort et al., 2014). For each sample, 150 fields of view were automatically captured and analyzed with the SYRACO (SYstème de Reconnaissance Automatique de Coccolithes) software at the European Centre for Research and Teaching in Environmental Geosciences (CEREGE, France), yielding counts ranging from 1,565 to 20,443 coccolith specimens (average ~6,219 per sample). SYRACO applies neural network

algorithms to classify coccoliths and to extract quantitative parameters from segmented digital images, enabling high-throughput, and standardised assessment of coccolith assemblages (Beaufort et al., 2014).


Water column stratification was assessed using a modified version of the *Florisphaera profunda* index for the Pleistocene Indian Ocean record (Beaufort et al., 1997; Beaufort et al., 2001). To better reflect the assemblage dynamics observed in our Pliocene record, we adapted the original formula by using the relative abundances of small *Gephyrocapsa* (<3 μm) and *Reticulofenestra* (<3 μm, 3-5 μm), which exhibit ecological behaviour similar to the late Pleistocene species *Emiliania*

*huxleyi* as used in the original equation. *F. profunda* is a deep photic zone species (~60 to 200 m in the water column; Winter et al., 1994), typically dominant under conditions of a deep thermocline and/or nutricline and low surface water productivity (Molfino and Mcintyre, 1990). In contrast, *Gephyrocapsa* and *Reticulofenestra* are characteristic of the upper photic zone. Consequently, lower values of the stratification index indicate a more mixed water column, whereas values approaching 1 reflect stronger stratification. The modified stratification index is calculated as:


$$F.\ profunda\ \text{index} = \frac{F.\ profunda}{(F.\ profunda + \text{small}\ Reticulofenestra + Reticulofenestra\ 3-5\ \mu m + Gephyrocapsa\ <3\ \mu m)} \qquad (1)$$

## 2.8 Statistical and time series analyses

To identify significant frequencies in the $\delta^{13}$C and $\delta^{18}$O isotopic signals, we performed single-Spectral analysis using the

Acycle software package (Li et al., 2019). Prior to analysis, the stable isotope time series were linearly interpolated to equally spaced 2.25 kyr time intervals. Bandpass filtering was applied to isolate specific orbital-scale components using the Astrochron R package (Meyers, 2014). Additionally, Morlet wavelet analysis was performed using the Paleontological Statistics software package (PAST; Hammer et al., 2009) to investigate the temporal evolution of periodicities in the time series.

**3 Results and discussion**

## 3.1 Vertical water column plankton community structure

The $\delta^{13}$C record from the benthic foraminifera *C. wuellerstorfi* ($\delta^{13}$C$_{BF}$) exhibits values ranging from −0.40 to +0.82 ‰, with a mean of +0.35 ‰ (**Fig. 2b**). The overall positive values imply that the deep waters at Site U1476 were moderately enriched in $^{13}$C, indicative of either recent ventilation or limited accumulation of respired carbon (Kroopnick, 1985). The amplitude of

variability of 1.22 ‰ is comparable to that of other mPWP deep-water records in the Atlantic and Pacific, suggesting that while deep-water mass properties at Site U1476 were dynamic, they did not undergo major reorganisation or shift in water mass structure (e.g., Novak et al., 2024; Braaten et al., 2023).

The $\delta^{13}C$ record from the planktic foraminifera *G. ruber* ($\delta^{13}C_{PF}$) ranges from +0.52 to +1.63 ‰ (mean +1.10 ‰; **Fig. 2c**). These values align with the well-documented habitat of *G. ruber*, a surface-dwelling taxon that calcifies in the upper mixed layer (20-50 m; Mohtadi et al., 2011) and thus reflect the uppermost ocean $\delta^{13}C$ of DIC. This $\delta^{13}C$ signal is influenced by photosynthetic carbon fixation and air-sea gas exchange (e.g., Ziveri et al., 2007; Bijma et al., 1999). The consistently positive $\delta^{13}C$ values throughout the interval suggest a continuous biological drawdown of $^{12}C$ and an active surface-ocean biological pump, most likely supported by nutrient delivery to the photic zone via localised upwelling (e.g., Faul et al., 2000) or thermocline shoaling events (Fedorov et al., 2013).

In contrast, the coccolith fraction $\delta^{13}C$ ($\delta^{13}C_{CO}$) record exhibits substantially lower values than *G. ruber*, ranging from −0.49 to +0.74 ‰, with an average of +0.09 ‰ (**Fig. 2d**). Coccolithophores (calcareous phytoplankton) are primary producers inhabiting the photic zone (<200 m), often residing higher in the water column than planktic foraminifera (Ziveri et al., 2007). Still, some exceptional species, such as *F. profunda*, are found deeper within the photic zone (Winter et al., 1994) (**Fig. 2a**). Interestingly, the range of $\delta^{13}C_{CO}$ values overlaps with that of $\delta^{13}C_{BF}$ at this site. However, $\delta^{13}C_{CO}$ shows greater variability (**Fig. 2d**), with an overall trend similar to $\delta^{13}C_{PF}$ (**Fig. 2c**). This similarity in the range of $\delta^{13}C_{CO}$ values with the benthic record may indicate a partial integration of deep photic zone DIC signals, especially under stratified conditions. This occurs because the respiration of organic matter at depth within an isolated deep photic zone creates a reservoir of $^{12}C$-enriched DIC (Kroopnick, 1985). While benthic foraminifera record the $\delta^{13}C$ of well-ventilated deep waters, the coccolith fraction can record this $^{13}C$-depleted respired carbon signature from the lower photic zone, leading to values that overlap with or are lower than the benthic record. The greater variability in $\delta^{13}C_{CO}$ likely results from a combination of mixed species assemblages, ecological variability in coccolithophore depth habitats, and differing physiological controls on carbon isotopic fractionation (Ziveri et al., 2003; Stoll et al., 2007a; Bolton et al., 2012).

In general, $\delta^{13}C_{CO}$ values would be predicted to be higher compared to deeper calcifiers, if directly linked to photosynthetic fractionation. However, the lower average $\delta^{13}C_{CO}$ relative to $\delta^{13}C_{PF}$ at Site U1476 implies the influence of isotopically light DIC, likely originating from recycled organic carbon or stratified surface layers with limited vertical exchange (Stirnimann et al., 2024). In addition to water-column dynamics, $\delta^{13}C_{CO}$ is influenced by species-specific vital effects—biologically mediated isotopic fractionations arising from internal cellular processes during calcification (Ziveri et al., 2003; Stoll et al., 2002). These effects can vary considerably among taxa due to differences in carbon-acquisition mechanisms, cellular carbon demand, and calcification rates (Bolton et al., 2012). Such a discrepancy may also reflect seasonal or ecological variations in calcification depth, or a decoupling of coccolithophore and planktic foraminifera habitats, resulting from differential nutrient availability or varying growth seasons (e.g., Jonkers and Kučera, 2017; Stoll et al., 2007b). This hypothesis is supported by the high relative abundances of the deep photic zone coccolithophore species *F. profunda* (**Fig. 2f**). Its abundance here denotes a strongly stratified upper ocean and suggests that a substantial portion of coccolith production likely occurred in deeper, nutrient-transition layers where the $\delta^{13}C$ of DIC is more heavily influenced by remineralised organic matter,

potentially accounting for the overall low $\delta^{13}C_{CO}$ values. Although species-specific $\delta^{13}C$ measurements for *F. profunda* have not been documented, its ecological niche, being adapted to low-light, nutrient-rich environments and potentially relying on carbon sources with greater respired DIC contributions, implies that its calcite may carry more negative $\delta^{13}C$ values compared to the flora in the upper photic zone. This species-specific signal, if present, could represent a vital effect contributing to the $\delta^{13}C_{CO}$–$\delta^{13}C_{PF}$ offset observed in the record.

The $\delta^{18}O$ values of benthic foraminifera ($\delta^{18}O_{BF}$) range from +2.09 to +3.28 ‰, with a mean value of +2.70 ‰ (**Fig. 2g**). These values are indicative of the cold and saline deep water characteristic of the Pliocene Indian Ocean, consistent with previous studies (e.g., Karas et al., 2009; Dupont et al., 2005; Hodell and Venz, 1992). In contrast, the $\delta^{18}O$ values of the planktic foraminifera *G. ruber* ($\delta^{18}O_{PF}$) exhibit much lighter values, ranging from −2.20 to +1.51 ‰, yielding an average of −1.84 ‰ (**Fig. 2h**). This record reflects warm SSTs and potentially reduced surface salinity, possibly linked to regional freshwater inputs or monsoonal variability (Saraswat et al., 2023) in the Mozambique Channel.

Coccolith $\delta^{18}O$ ($\delta^{18}O_{CO}$) values range from −1.39 to −0.64 ‰, with an average of −1.03 ‰ (**Fig. 2i**). Although coccolithophores are typically photosynthetic and inhabit the upper photic zone, the heavier $\delta^{18}O_{CO}$ compared to *G. ruber* suggests that a significant portion of this community's coccolith calcification may have occurred deeper within the photic zone. This interpretation assumes that both coccolithophores and *G. ruber* record similar seasonal conditions and reflect the $\delta^{18}O$ of ambient seawater without significant species-specific vital-effect offsets on $\delta^{18}O$. Consistent with the $\delta^{13}C_{CO}$ recorded here, this contradicts the notion that coccolithophores calcify in shallower waters than *G. ruber*, which generally resides in the upper mixed layer. The heavier $\delta^{18}O_{CO}$ values could indicate significant contributions from deeper-dwelling coccolithophore species, especially those inhabiting the lower photic zone (up to ~200 m), such as *F. profunda* (Winter et al., 1994), supporting the interpretation that the coccolith fraction integrates signals from a broader vertical range of the photic zone, including thermally and isotopically distinct layers. Accordingly, the vertical gradient observed in $\delta^{18}O$, from heavier values in benthic foraminifera, to intermediate values in coccoliths, and the lightest values in *G. ruber*, indicates a stratified ocean structure, where $\delta^{18}O_{CO}$ represents conditions within the mid to lower photic zone, potentially influenced by deeper-dwelling species rather than solely the surface mixed layer. These differences in $\delta^{13}C$ and $\delta^{18}O$ values between the three plankton groups thus offer valuable insights into the vertical structure of the Pliocene water column and help constrain the ecological depth habitat of these plankton communities.

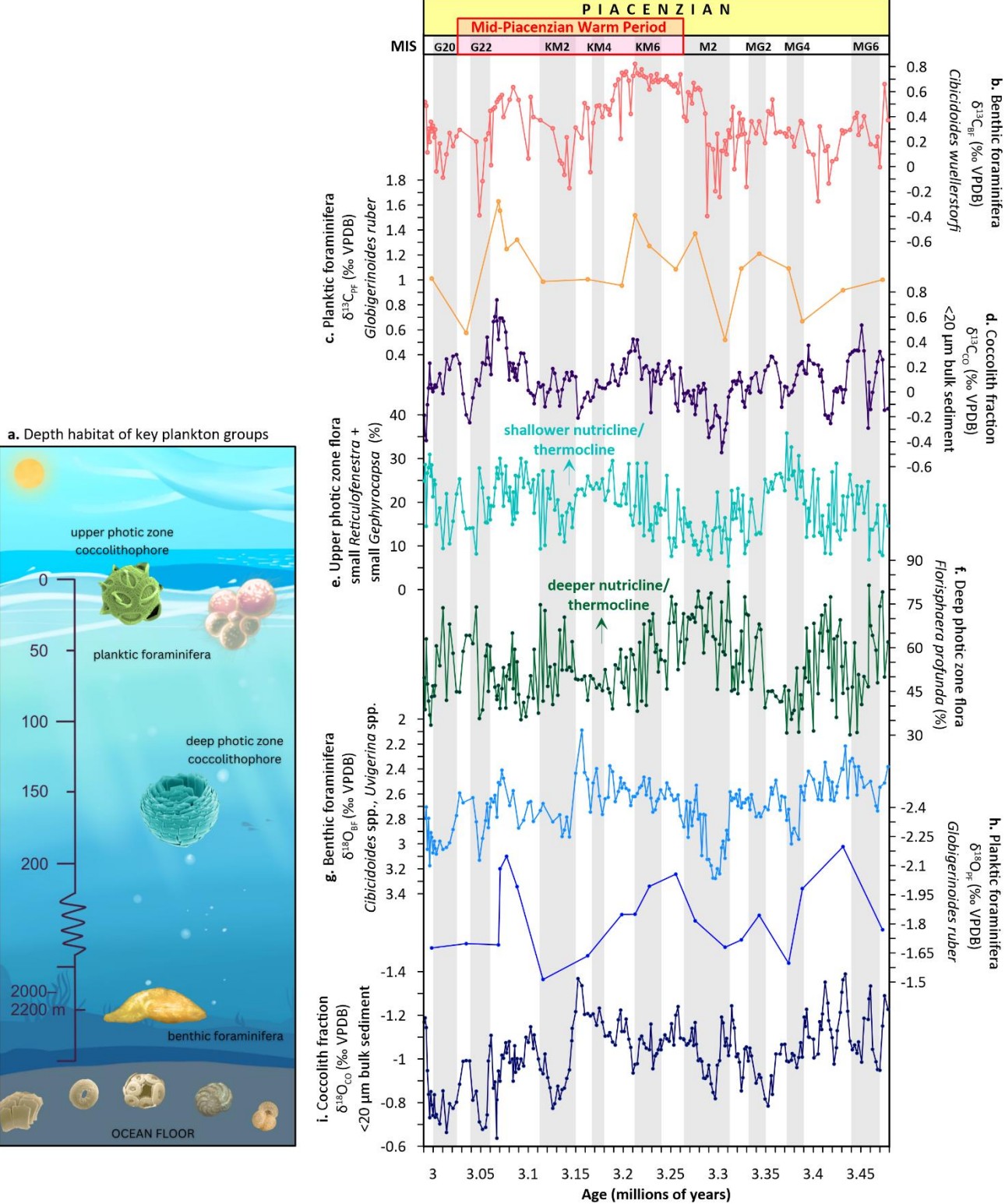

**Figure 2: Carbon and oxygen isotope records and coccolithophore abundance data from depth-stratified plankton groups at Site U1476. (a) depth habitat of key plankton groups (microfossil images, http://www.mikrotax.org/Nannotax3), (b) benthic foraminifera *Cibicidoides wuellerstorfi* $\delta^{13}C_{BF}$, (c) planktic foraminifera *Globigerinoides ruber* $\delta^{13}C_{PF}$, (d) coccolith fraction $\delta^{13}C_{CO}$ (<20 µm), (e) upper photic zone flora (small *Gephyrocapsa* and small *Reticulofenestra*) relative abundance, (f) deep photic zone flora *Florisphaera profunda* relative abundance, (g) benthic foraminifera *C. wuellerstorfi*, *Cibicidoides* spp., *Uvigerina* spp. $\delta^{18}O_{BF}$, (h) planktic foraminifera *G. ruber* $\delta^{18}O_{PF}$, (i) coccolith fraction $\delta^{18}O_{CO}$ (<20 µm). Glacial stages are shown by grey horizontal bars (Lisiecki and Raymo, 2005).**

## 3.2 Interpreting the $\delta^{13}C$ and $\delta^{18}O$ signature of coccolithophores relative to *G. ruber*

The $\delta^{13}C$ and $\delta^{18}O$ values derived from coccoliths are consistently more negative than those of the surface-dwelling planktic foraminifera *G. ruber* (**Fig. 2c–d, 2h-i, S2d–g**) despite both groups calcifying within the photic zone. Here we address these isotopic offsets.

Coccolithophore assemblage data from Site U1476 show that the community in this study area is strongly dominated by the deep-dwelling taxon *F. profunda* (**Fig. S2a**). Consequently, its dominance can shift the isotopic signal of the coccolith fraction downward, particularly under stable stratification where production is high in the lower photic zone. Morphometric analysis using SYRACO shows that despite its consistently high relative abundances (**Fig. 2f, S2a**), *F. profunda* contributes only up to 2 % of the total coccolith calcite mass (**Fig. S2b**). In contrast, larger and more heavily calcified taxa such as *Reticulofenestra* spp. (up to ~43 %), *Pontosphaera* spp. (up to 39 %), *Helicosphaera* spp. (up to ~26 %), and *Calcidiscus leptoporus* (up to ~23 %) dominate the total calcite mass. However, when considering the total calcium carbonate (CaCO₃) contribution, *F. profunda* accounts for 11–69 % (mean ~34 %) of the total CaCO₃ in the <20 µm fraction compared to *Reticulofenestra* spp. (20–67 %, mean ~43 %), with each of the remaining taxa contributing <7 % on average (**Fig. S2c**).

This distinction between the relative abundance and calcite mass is critical. While *F. profunda* coccoliths numerically dominate the assemblages, their thinner coccoliths reduce their mass contribution. Nonetheless, its consistent presence and substantial share of the CaCO₃ budget (**Fig. S2c**), can still skew the coccolith fraction isotopic signal toward lower values in the photic zone. This likely contributes to the more negative $\delta^{13}C$ and $\delta^{18}O$ values of the coccolith fraction relative to *G. ruber*, which calcifies in the well-lit, surface mixed layer and better reflects surface $\delta^{13}C$-DIC and SSTs (Mohtadi et al., 2011). Poulton et al. (2017) further support this interpretation, showing that *F. profunda* dominates during stratified seasons when the nutricline deepens, and suggest that its persistence in low-light, deep photic conditions, possibly through mixotrophy or phagotrophy, reflects calcification well below the surface layer.

Interestingly, $\delta^{13}C_{CO}$ values partially converge with $\delta^{13}C_{BF}$, despite originating from much shallower depths. This pattern suggests that coccoliths may record DIC signals from the deep photic zone, where $\delta^{13}C$ can resemble that of intermediate or deep waters due to vertical segregation and respiration-driven $^{12}C$ enrichment (Broecker and Maier-Reimer, 2012). While their value overlaps with $\delta^{13}C_{BF}$, the higher variability in $\delta^{13}C_{CO}$ more closely resembles that of *G. ruber*, reflecting a mixed signal shaped by seasonal productivity, species turnover, and changing ecological niches.

Vital effects further complicate the interpretation of coccolith fraction isotopic records. Coccolithophores exhibit species-specific physiological controls on $\delta^{13}C$ and $\delta^{18}O$ fractionation, driven by differences in cellular carbon uptake, internal $CO_2$ concentration, and calcification rate, which can be attributed to a combination of ecological community structure, depth-related environmental gradients, and physiological (vital) effects (Bolton et al., 2012; Stoll, 2005; Ziveri et al., 2003; **Table S3**). Taxa such as *Helicosphaera* and *Calcidiscus* are known to display large $\delta^{13}C$ offsets even when cultured under identical conditions (Ziveri et al., 2003). The coccolith fraction isotopic signal thus reflects a weighted average of multiple species, each with its physiological imprint.

Another consideration is the potential for diagenetic overprinting. Unlike planktic foraminifera, where hand-picking ensures species purity, coccolith isotope data are derived from the <20 µm fine fraction bulk carbonate, which may include other fine carbonate phases (e.g., foraminifera and calcareous dinoflagellate fragments). Nevertheless, several lines of evidence suggest that the signal is mainly of primary in origin. First, the sieving and processing protocol minimizes the inclusion of foraminiferal fragments and coarse recrystallized grains. Second, smear-slide inspection under light microscopy shows that the <20 µm fraction is almost entirely composed of coccoliths (≥95 % by visual estimation), with only trace or no occurrences of other very fine-grained carbonate particles. The coccoliths display well-preserved, intact shields with no visible secondary calcite overgrowth, dissolution pitting, or micritic cement, all of which would indicate diagenetic alteration. Third, the isotopic trends observed (e.g., depleted $\delta^{13}C$ in association with high *F. profunda*) are internally consistent with known ecological and depth preferences (Winter et al., 1994).

Given these complexities, caution is warranted when interpreting coccolith isotopic signals in bulk sediments. The influence of dominant taxa, such as *F. profunda*, combined with species-specific vital effects and depth-integrated geochemical gradients, may obscure signals from shallower-dwelling forms. This highlights the need for species-specific geochemical analyses to resolve the contributions of individual taxa to the coccolith fraction signal. *F. profunda* in particular, with its association with oligotrophic environments and apparent ability to leave a disproportionate imprint on the carbonate record, presents a compelling target for future research aimed at refining palaeoceanographic reconstructions in stratified tropical settings.

## 3.3 Carbon cycling and stratification across the Piacenzian

High-resolution $\delta^{13}C$ and $\delta^{18}O$ records from Site U1476 spanning 3.48–2.99 Ma reveal sustained vertical isotopic gradients between surface (coccolithophores and planktic foraminifera) and deep (benthic foraminifera) carbon reservoirs during the Piacenzian (**Fig. 3; Table S4**). These gradients offer critical insights for understanding vertical ocean structure, water mass stratification, and the efficiency of the biological pump in the Mozambique Channel during a climate interval marked by both elevated global temperatures and moderate $p$CO$_2$ levels.

From 3.48 to ~3.31 Ma (prior to MIS M2 glaciation), the benthic foraminifera–coccolith $\delta^{13}C$ gradient ($\Delta\delta^{13}C_{BF-CO}$) remained relatively stable (**Fig. 3a**), suggesting sustained separation between surface and deep carbon reservoirs, indicative of a moderately efficient biological pump. A notable exception is a transient decline in $\Delta\delta^{13}C_{BF-CO}$ and the planktic foraminifera–coccolith $\delta^{13}C$ gradient ($\Delta\delta^{13}C_{PF-CO}$; **Fig. 3c**) between ~3.42 and 3.39 Ma, suggesting a brief episode of increased deep-ocean ventilation (**Fig. 3a**) and surface ocean mixing (**Fig. 3c**). Interestingly, during this same interval, the *F. profunda* stratification index briefly increases, typically signifying a well-developed deep chlorophyll maximum (e.g., Molfino and Mcintyre, 1990) and strong upper ocean stratification (e.g., Beaufort et al., 2001) (**Fig. 3d**). This suggests that while surface and subsurface layers remained stratified, limiting nutrient entrainment into the euphotic zone, ventilation and mixing occurred primarily at intermediate depths, sufficient to weaken deep–surface isotopic offsets without entirely disrupting the upper ocean density structure. Thus, the ocean likely experienced subsurface ventilation beneath a still-stratified surface layer, reflecting vertical decoupling between surface productivity dynamics and deeper overturning processes.

A progressive increase in $\Delta\delta^{13}C$ gradients occurred approaching MIS M2 glaciation, with $\Delta\delta^{13}C_{BF-CO}$ rising by +1.35 ‰ (**Fig. 3a**) and $\Delta\delta^{13}C_{PF-CO}$ by +1.13 ‰ (**Fig. 3c**). This increase in gradients, along with a sustained rise in the *F. profunda* index (**Fig. 3d**), suggests a re-establishment of a stratified ocean with reduced vertical exchange, consistent with long-term warming and decreased deep-water renewal (e.g., Lisiecki and Raymo, 2005; Andersson et al., 2002). During this phase, the benthic–planktic foraminifera $\delta^{13}C$ gradient ($\Delta\delta^{13}C_{BF-PF}$) remained moderately negative (~–0.73 ‰; **Fig. 3b**), reflecting the remineralization gradient within the mesopelagic zone (Hodell and Venz-Curtis, 2006), and supporting relatively stable export production processes.

The MIS M2 glaciation (~3.30–3.28 Ma) marks a significant departure from preceding conditions. At Site U1476, it is characterised by a sharp increase in the benthic foraminifera–coccolith $\delta^{18}O$ gradient ($\Delta\delta^{18}O_{BF-CO}$) of +0.78 ‰ (equivalent to ~–3.36 °C cooling) (**Fig. 3e**), and a peak in the benthic–planktic foraminifera $\delta^{18}O$ gradient ($\Delta\delta^{18}O_{BF-PF}$) (**Fig. 3f**), indicative of deep ocean cooling and enhanced vertical thermal stratification (De Schepper et al., 2014; De Schepper et al., 2009). Simultaneously, the *F. profunda* index reaches a sustained peak, signalling intensified surface ocean stratification, supported by a peak in the planktic foraminifera–coccolith $\delta^{18}O$ gradient ($\Delta\delta^{18}O_{PF-CO}$) (**Fig. 3g**). This points to a

strengthened upper-ocean nutrient trap and reduced mixing, consistent with suppressed AMOC during M2 (Bell et al., 2015).
These findings align with SST cooling observed in tropical and subtropical oceans (Herbert et al., 2016; McClymont et al., 2020) and support reduced vertical heat transport during M2 (Zhang et al., 2021). PlioMIP Phase 2 simulations further suggest cooler conditions during M2, with expanded sea ice and regional cooling, particularly in high-latitude regions (e.g., Dolan et al., 2015; Haywood et al., 2020), reflecting a transient but globally significant glaciation within the generally warm mid-Pliocene climate.


The mPWP (~3.264–3.025 Ma) is characterised by overall stable vertical $\delta^{13}C$ gradients, relative to pre-M2 conditions. Although this interval was globally marked by high SSTs (~2–3 °C above modern; McClymont et al., 2023) and elevated $pCO_2$ (~350–450 ppm; Seki et al., 2010; de la Vega et al., 2020), Site U1476 reflects a regime of strong thermal stratification but reduced isotopic separation. Assemblage data indicate a proliferation of upper photic zone coccolithophore species (**Fig. 2e**), consistent with enhanced surface-layer productivity. However, the stable $\delta^{13}C$ gradients and limited deep-water ventilation suggest that this productivity was primarily confined to the nutrient-replete surface and did not translate into efficient export to the deep ocean. This is consistent with a suppressed deep nutrient supply during this period due to reduced overturning and weakened thermocline ventilation, a pattern also identified in the North Pacific during the mid-Pliocene (Ford et al., 2022), and supported by proxy-model comparisons showing reduced equatorial thermocline ventilation and
increased ocean heat content under mid-Pliocene warmth (Ford et al., 2025). PlioMIP models simulate this stable warm period with reduced AMOC strength and weakened equatorial upwelling, particularly in the Indo-Pacific (Zhang et al., 2021; Haywood et al., 2020), which would have promoted nutrient depletion below the surface and limited biological carbon export despite high surface productivity. The apparent decoupling between elevated surface productivity and stable vertical $\delta^{13}C$ gradients during the mPWP likely reflects nutrient recycling within the surface layer under strong stratification rather
than enhanced export to depth. However, as this inference is derived solely from isotopic and assemblage patterns without supporting geochemical tracers of export flux (e.g., opal, Ba/Al, organic carbon), it should be considered a first-order interpretation subject to validation by additional proxy records.

During the mPWP, a sharp collapse in vertical $\delta^{13}C$ gradients during MIS KM2 is observed: $\Delta\delta^{13}C_{BF-CO}$ declines by –1.03 ‰
(**Fig. 3a**), and $\Delta\delta^{13}C_{BF-PF}$ by –0.89 ‰ (**Fig. 3b**), and $\Delta\delta^{13}C_{PF-CO}$ by –0.78 ‰ (**Fig. 3c**). These shifts indicate a breakdown in vertical carbon reservoir separation and suggest a pulse of enhanced ventilation, consistent with Northern Hemisphere glaciation and global cooling events documented in other records (De Schepper et al., 2014; McClymont et al., 2023). In parallel, $\delta^{18}O$ gradients decline: the $\Delta\delta^{18}O_{BF-CO}$ decreases by –0.94 ‰ (equivalent to +4.05 °C warming; **Fig. 3e**), the benthic–planktic foraminifera $\delta^{18}O$ gradient ($\Delta\delta^{18}O_{BF-PF}$) by –0.60 ‰ (+2.59 °C warming; **Fig. 3f**), and the $\Delta\delta^{18}O_{PF-CO}$ by –
0.33 ‰ (+1.43 °C warming; **Fig. 3g**), collectively pointing to surface and subsurface warming and a deepening of the thermocline. However, the *F. profunda* index declines during this period, supporting enhanced surface mixing, possibly from lateral advection of nutrients carried by the East African Coastal Current (Tangunan et al., 2017), which may have supported

sustained productivity and an increase in the upper photic species (**Fig. 2e**), despite an overall background of stratified conditions (Lawrence et al., 2006).


The rest of the mPWP saw amplified variability in $\Delta \delta^{13}C_{BF-CO}$ (**Fig. 3a**) and $\Delta \delta^{13}C_{BF-PF}$ (+0.78 ‰; **Fig. 3b**), indicating enhanced coupling between surface and deep waters and a modest reinvigoration of the biological pump (Martínez-Méndez et al., 2010). This is also reflected in the *F. profunda* index (**Fig. 3b**), which displays high variability, signifying dynamic shifts in nutricline depth and surface ocean productivity. The $\Delta \delta^{18}O_{BF-CO}$ (**Fig. 3e**) shows pronounced variability, suggesting

recurrent reorganisations of deep-water masses (e.g., Hodell and Venz-Curtis, 2006). Maxima in isotopic gradients for both $\Delta \delta^{18}O_{BF-CO}$ (**Fig. 3e**) and $\Delta \delta^{18}O_{BF-PF}$ (**Fig. 3f**) suggest strengthened vertical thermal gradients while minima in $\Delta \delta^{18}O_{BF-CO}$ (**Fig. 3e**) and $\Delta \delta^{18}O_{PF-CO}$ (**Fig. 3g**) point to episodes of upper-ocean warming or freshwater input likely linked to changes in the precipitation-evaporation balance (Brierley and Fedorov, 2016). However, the planktic foraminifera record is a lower-resolution archive relative to the coccolith and benthic foraminifera records, potentially limiting its fidelity in capturing

short-term $\delta^{13}C$ and $\delta^{18}O$ variability. These changes precede the intensification of Northern Hemisphere glaciation and may signal the onset of more dynamic ocean circulation regimes (Lisiecki and Raymo, 2005), foreshadowing the Pleistocene transition and the establishment of the Northern Hemisphere glaciation (~2.7 Ma).

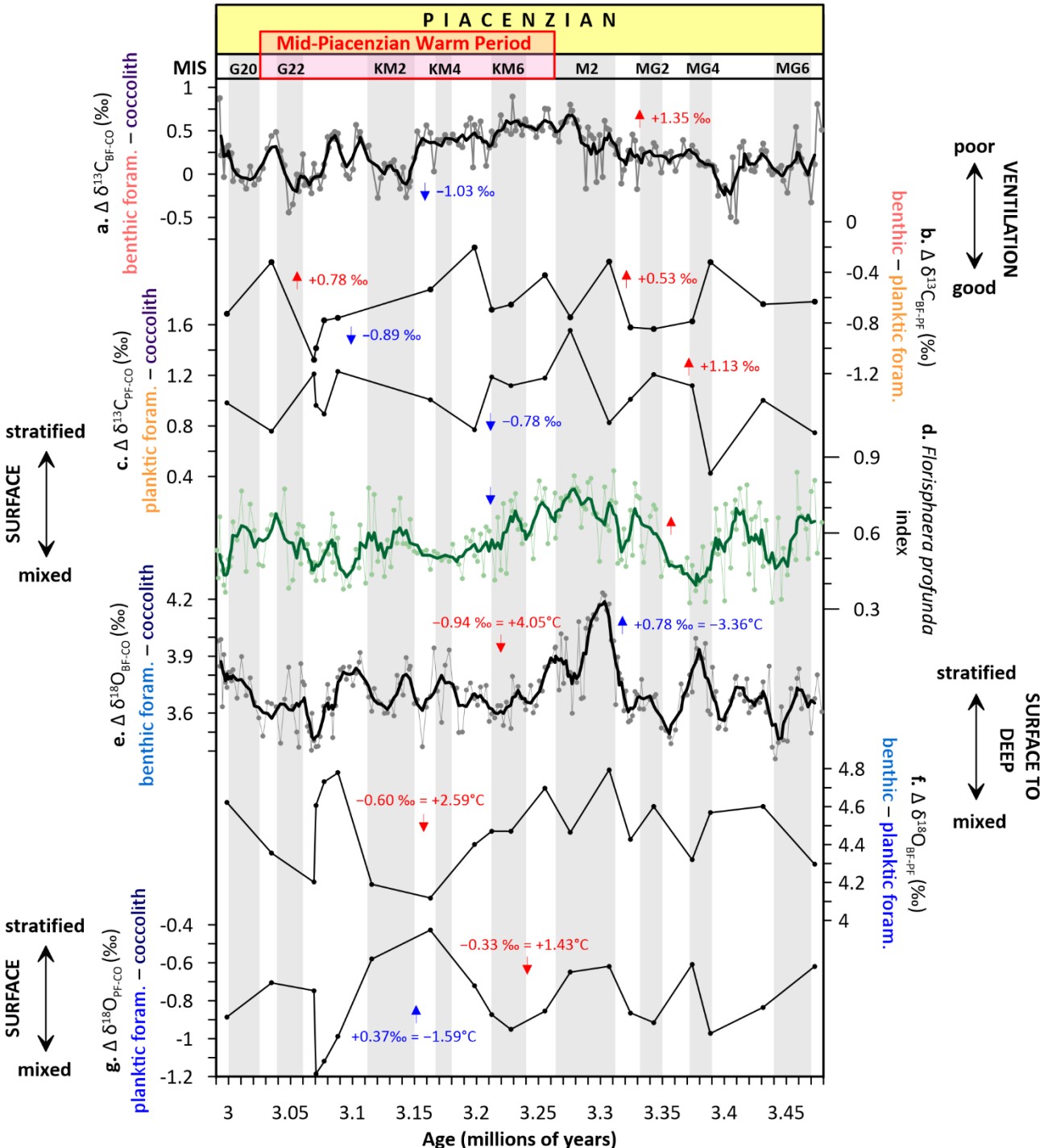

**Figure 3: Carbon and oxygen isotopic gradients (Δ) at Site U1476. (a) benthic foraminifera (BF)−coccolith fraction (CO) δ¹³C, (b) benthic−planktic foraminifera (PF) δ¹³C, (c) coccolith fraction−planktic foraminifera δ¹³C, (d) *Florisphaera profunda* stratification index calculated as the ratio of the deep photic zone taxon *F. profunda* and the combined abundances of upper photic zone taxa: small Gephyrocapsa, small *Reticulofenestra*, and *Reticulofenestra* 3-5 μm, (e) benthic foraminifera−coccolith fraction δ¹⁸O, (f) benthic−planktic foraminifera δ¹⁸O, (g) planktic foraminifera−coccolith fraction δ¹⁸O. Labelled arrows highlight major long-term trends superimposed on glacial–interglacial variability. Glacial stages are shown by grey horizontal bars (Lisiecki and Raymo, 2005).**

## 3.4 Orbital scale controls on Pliocene ocean-climate dynamics

While the Pliocene is often characterised as a climatically stable interval with globally elevated temperatures and moderate $p$CO$_2$ levels, recent studies underscore the pivotal role of orbital forcing in modulating both surface and deep ocean conditions during this epoch (e.g., De Schepper et al., 2014; McClymont et al., 2020). Variations in eccentricity, obliquity, and precession have been shown to exert strong control over insolation patterns, which in turn drive changes in ocean circulation, carbon storage, and ice sheet dynamics (Lisiecki and Raymo, 2005). Particularly during the mPWP, climate responses to orbital-scale forcing were regionally diverse and temporally complex, often mediated by ocean-atmosphere feedback and changes in the AMOC (e.g., Brierley et al., 2009; Fedorov et al., 2013). Here we examine the imprint of orbital pacing on δ¹³C and δ¹⁸O variability, revealing how eccentricity-modulated obliquity and precession rhythms governed the ventilation state of the deep ocean and the efficiency of surface–deep carbon exchange across the Piacenzian.

Our high-resolution coccolith and benthic foraminifera δ¹³C and δ¹⁸O records from IODP Site U1476 reveal distinct orbital pacing of water column structure during the Piacenzian (**Fig. 4, 5**). The δ¹³C$_{BF}$ record shows strong eccentricity (~100 kyr) variability (**Fig. 4a–c**), indicating that deep ocean carbon storage responded fundamentally to long-term orbital variations. This likely reflects eccentricity-modulated changes in Southern Ocean ventilation that altered the injection of ¹³C-enriched waters into the deep Indian Ocean (Hodell and Venz-Curtis, 2006), coupled with potential variations in organic carbon burial efficiency (Sarmiento and Gruber, 2013). While eccentricity dominates the deep carbon cycle, obliquity (~41 kyr) exerts strong control on deep ocean temperature (δ¹⁸O$_{BF}$; **Fig. 4d–f**) and photic zone carbon cycling (δ¹³C$_{CO}$; **Fig. 4g–i**). This obliquity signal reflects the high-latitude forcing of Southern Ocean ventilation (Hodell and Venz-Curtis, 2006) and associated changes in Indian Ocean circulation (Biastoch et al., 2009) (**Fig. 5**), consistent with the global benthic isotope stack (Lisiecki and Raymo, 2005).

Notably, precession (~19–23 kyr) dominates surface hydrography (δ¹⁸O$_{CO}$; **Fig. 4j–l**), especially prior to MIS M2 during eccentricity maxima, with peaks coinciding with summer insolation maxima at Site U1476 (15° S; **Fig. 5a**) and the Southern Hemisphere (65° S), and the orbital precession expressed in the normalized ETP solution (Laskar et al., 2011; **Fig. 5b**). This tropical-subtropical response highlights the dual sensitivity of the Mozambique Channel to both high latitude forcing via

Southern Hemisphere westerlies (Toggweiler et al., 2006; Biastoch et al., 2009) and low latitude forcing via Indian monsoon dynamics (Clemens et al., 1991) and the Indo-Pacific Walker Circulation (Van Der Lubbe et al., 2021). As a critical conduit for Indian-Atlantic Ocean exchange (Beal et al., 2011), the region's stratification and carbon storage responded dynamically to the interplay of these orbital forcings, driven by changes in the Southern Hemisphere westerlies (Biastoch et al., 2009),

Agulhas leakage (Schott et al., 2009; De Ruijter et al., 2002), and Indian-Atlantic circulation (De Ruijter et al., 2002; Beal et al., 2011).

Since the Site U1476 benthic $\delta^{18}O$ record is tuned to the LR04 global stack, we acknowledge that tie-point uncertainties of approximately ±3 to 5 kyr may affect the precise alignment of $\delta^{13}C$ and $\delta^{18}O$ signals with individual insolation peaks. This

uncertainty reflects both the ~2.25 kyr resolution of our isotope sampling and the correlation accuracy achievable during visual and statistical tuning to LR04 at orbital timescales (Lisiecki and Raymo, 2005; Meyers, 2014). While such uncertainties may influence the exact timing of proxy responses relative to insolation maxima or minima, they do not compromise the identification of obliquity- and precession-scale periodicities in our spectral and wavelet analyses. Accordingly, our interpretations emphasize the pacing and amplitude of orbital-scale variability rather than precise phase

relationships among orbital parameters, $p$CO$_2$, and proxy records. We did not attempt to resolve detailed lead–lag relationships, as our focus lies in characterizing the expression and pacing of orbital-scale variability in carbon cycling and stratification. The lead–lag patterns discussed later (**Section 3.5**) provide a qualitative comparisons with the independently derived $\delta^{11}B$-based $p$CO$_2$ record from ODP Site 999 (de la Vega et al., 2020), which explicitly resolved orbital-scale $p$CO$_2$ variability during the mid-Piacenzian. Our aim is to contextualise the Site U1476 isotope trends within that global CO$_2$

framework, rather than to infer mechanistic phase relationships. Thus, while age-model uncertainties may affect the precision of phase estimates, they do not undermine the broader interpretation of orbital forcing and its expression in the tropical Indian Ocean.

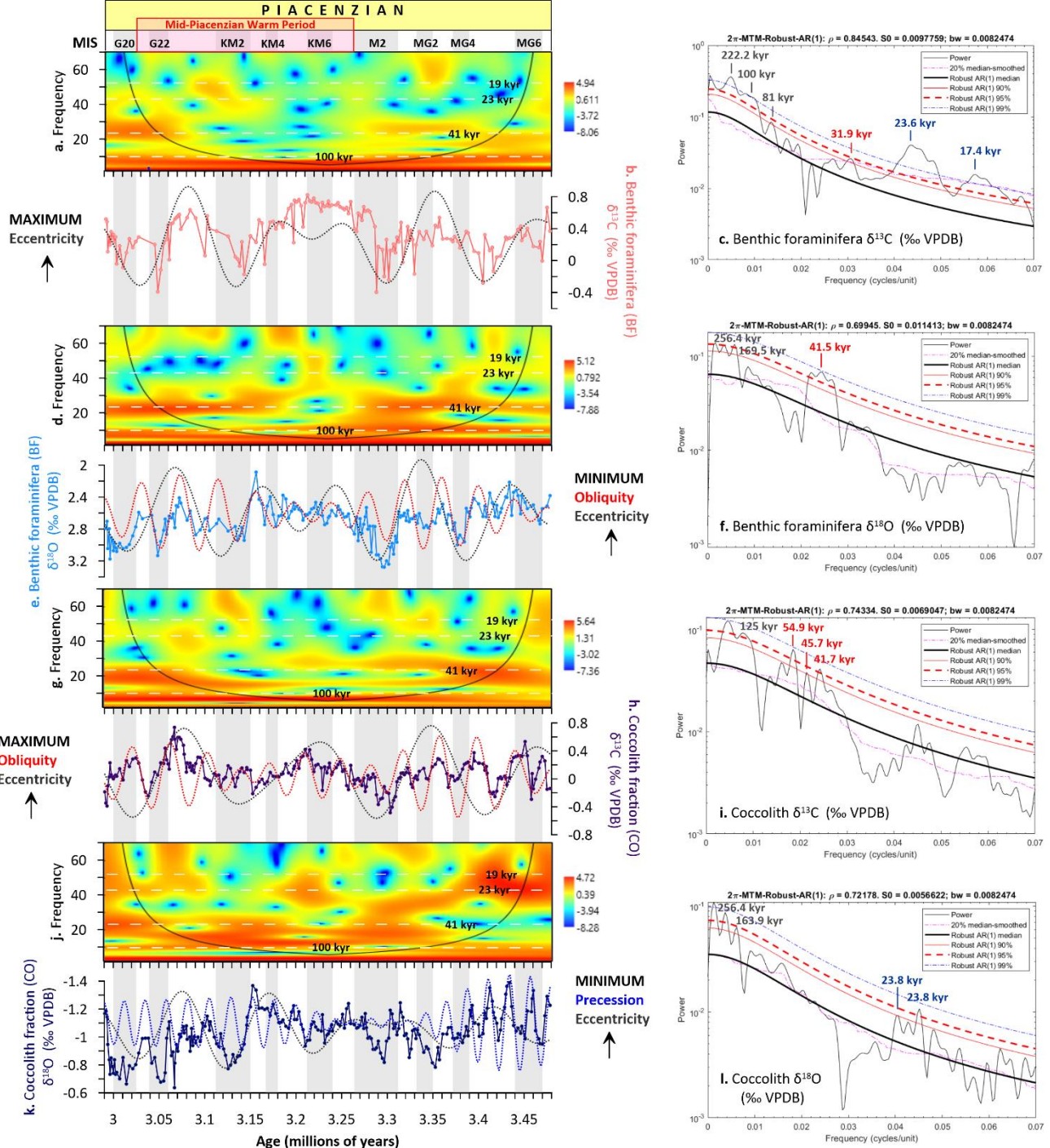

**Figure 4: Bandpass filtering and wavelet analysis of carbon and oxygen isotope records at Site U1476. Benthic foraminifera $\delta^{13}$C (a) wavelet power spectrum, (b) raw data with 100-kyr eccentricity filter, (c) multitaper method (MTM) power spectrum; benthic $\delta^{18}$O (d) wavelet power spectrum, (e) raw data with 100-kyr and 41-kyr obliquity filters, (f) MTM power spectrum; coccolith fraction $\delta^{13}$C (<20 μm) (g) wavelet power spectrum, (h) raw data with 100-kyr and 41-kyr filters, (i) MTM power spectrum; coccolith fraction $\delta^{18}$O (<20 μm) (j) wavelet power spectrum, (k) raw data with 100-kyr and 19–23 kyr precession filters, (l) MTM**

**power spectrum. Eccentricity = dotted black line, obliquity = dotted red line, precession = dotted blue line. The orbital target curves for filtering (eccentricity, obliquity, precession) are from Laskar et al. (2011). Glacial stages are shown by grey horizontal bars (Lisiecki and Raymo, 2005).**


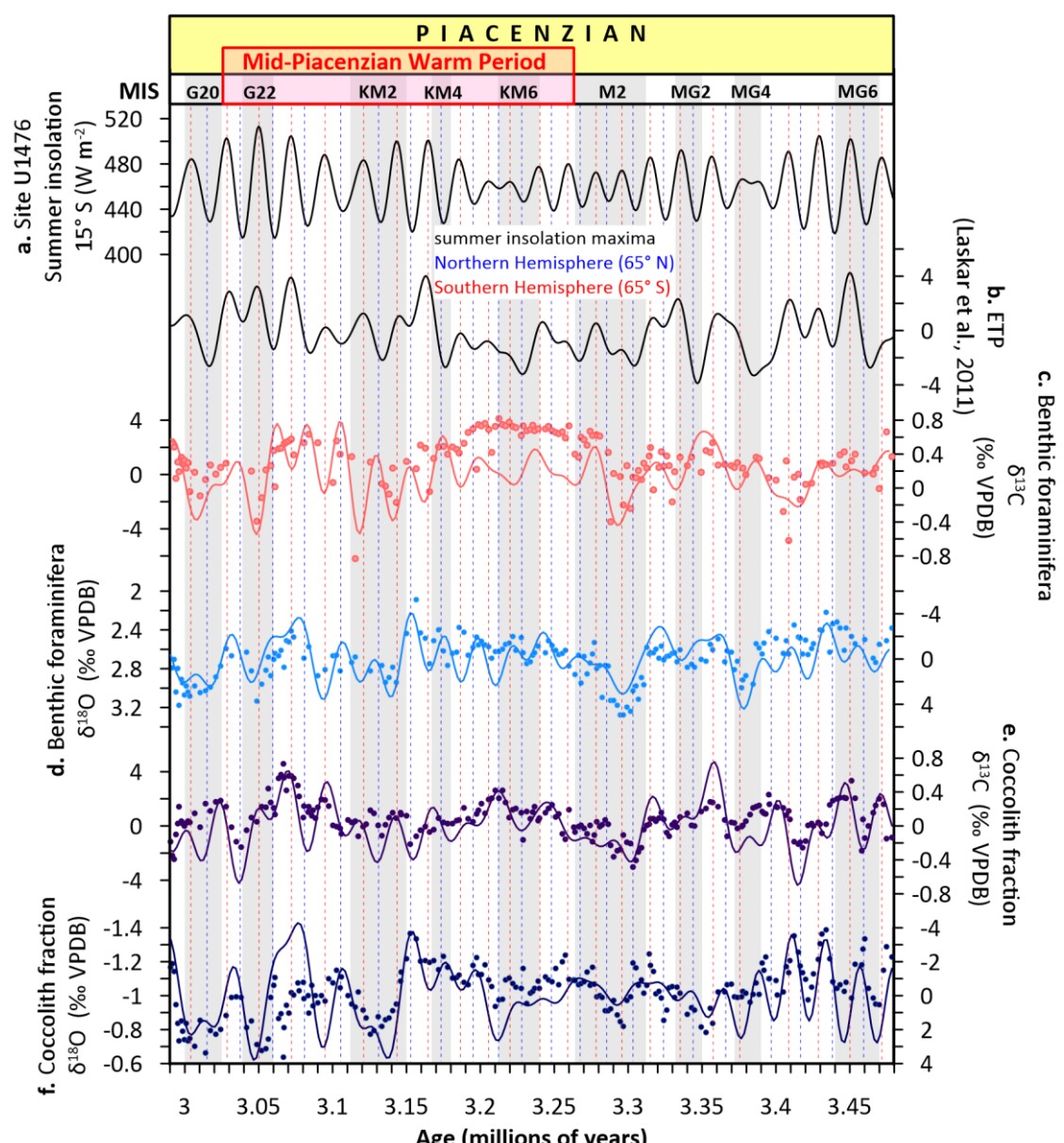

**Figure 5: Astronomical climate parameters and carbon and oxygen stable isotope records from Site U1476 during the Piacenzian. (a) summer insolation at 15°S (Site U1476), (b) ETP (eccentricity, obliquity/tilt, precession (Laskar et al., 2011), (c) benthic foraminifera δ¹³C, (d) benthic foraminifera δ¹⁸O, (e) coccolith fraction δ¹³C, (f) coccolith fraction δ¹⁸O. Solid lines in c–f represent ETP-filtered isotopic data. Blue dotted lines mark the peaks in δ¹³C and δ¹⁸O aligned with the Northern Hemisphere (65° N) summer insolation maxima while the red dotted lines denote peaks coinciding with the Southern Hemisphere (65° S) summer insolation maxima. Glacial stages are shown by grey horizontal bars (Lisiecki and Raymo, 2005).**


**3.5 Regional feedback and global context in a warm, high CO₂ world**

The co-variation of $\delta^{13}C$ and $\delta^{18}O$ records with reconstructed $pCO_2$ across the Piacenzian underscores a tightly coupled ocean-atmosphere carbon cycle and climate system in a warm, high-CO₂ world (**Fig. 6**, **Fig. S1d, Table S4**). We compared our $\delta^{13}C$ and $\delta^{18}O$ records with the high-resolution $pCO_2$ reconstruction of de la Vega et al. (2020) at ODP Site 999 in the Caribbean Sea (12°44' N, 78°44' W; 2828 m water depth), ensuring that the age models used in both datasets are compatible to allow meaningful evaluation of the trends (**Fig. S1c–f**). Both benthic foraminifera and coccolith $\delta^{13}C$ and $\delta^{18}O$ records follow a trend generally similar to that of $pCO_2$, suggesting that carbon cycling and ocean temperatures were globally coherent and responsive to changes in $pCO_2$ levels. These empirical observations are broadly consistent with PlioMIP Phase 1 and Phase 2 model outputs, which simulate a warmer-than-modern global mean temperature of 2.7–4.0 °C under mid-Piacenzian boundary conditions with elevated $CO_2$, and exhibit globally coherent patterns of surface warming and weakened meridional temperature gradients (Haywood et al., 2020; Haywood et al., 2007).

During the MIS M2 cool interval just prior to the mPWP, the early phase is characterised by a significant increase in $pCO_2$ (~420 ppm; **Fig. 6c**), which coincides with declining $\delta^{13}C_{BF}$ and $\delta^{13}C_{CO}$ values in both benthic foraminifera (**Fig. 6b**) and coccolithophores (**Fig. 6a**). This relationship likely reflects reduced surface ocean productivity and weakening of the biological pump due to enhanced stratification, resulting in decreased export of $^{13}C$-enriched organic matter to the deep ocean (Hodell and Venz, 1992; Honisch et al., 2009). PlioMIP2 simulations similarly suggest that elevated $CO_2$ concentrations during the mPWP suppressed vertical ocean mixing and intensified upper ocean stratification, particularly in tropical and subtropical regions, thereby reinforcing a regional feedback that weakens biological carbon export (Zhang et al., 2021). In such conditions, increased $CO_2$ uptake into surface waters would also lower $\delta^{13}C$ in DIC, further contributing to the negative $\delta^{13}C$ excursion observed across taxa. The biological pump is expected to be sensitive to changes in both ocean stratification and plankton metabolic functioning (Rost and Riebesell, 2004). Elevated $CO_2$ can enhance ocean stratification, which tends to reduce nutrient supply to surface waters and potentially limit primary productivity, weakening the biological pump. However, at the organism level, $CO_2$-induced changes in metabolic rates, particularly among calcifying phytoplankton, can also affect carbon export efficiency. Recent studies suggest that elevated $CO_2$ may increase metabolic activity in some taxa, influencing remineralisation depths and carbon sequestration pathways (Boscolo-Galazzo et al., 2021; Crichton et al., 2023). The persistent vertical $\delta^{13}C$ gradients observed here likely reflect a balance between these competing processes, with a stratified yet biologically active upper ocean supporting effective carbon export to the deep sea despite elevated $pCO_2$ levels.

While the general co-variation between $\delta^{13}C$, $\delta^{18}O$, and $pCO_2$ is evident, there are notable deviations from this trend, particularly during MIS M2, where lags emerge between Site U1476 and ODP Site 999 records. In the early stages of MIS M2, reconstructed $pCO_2$ appears to lag behind the $\delta^{13}C_{BF}$ and $\delta^{13}C_{CO}$ minima, suggesting a delayed $pCO_2$ response to oceanographic and biological changes (**Fig. 6a–c**). Similarly, the $pCO_2$ shows a delayed increase relative to $\delta^{18}O_{BF}$ (**Fig. 6e**),

whereas $\delta^{18}O_{CO}$ (**Fig. 6d**) aligns more closely with $pCO_2$ trends. This lag structure has been previously noted by de la Vega et al. (2020), suggesting a carbon cycle-based explanation, potentially linked to Southern Ocean dynamics. Specifically, the tail end of the $pCO_2$ decline during MIS M2 appears to be out of phase with Northern Hemisphere insolation but more aligned with insolation changes at 65° S, offset by half a precession cycle, hinting at high-latitude control over $CO_2$ drawdown. The fact that North Atlantic εNd records exhibit similar phasing supports a deep-ocean ventilation mechanism (Lang et al., 2016). Our data from Site U1476 confirm that $\delta^{13}C$ values in both benthic foraminifera and coccolith records also lag the $pCO_2$ inflexion point, consistent with this interpretation and reinforcing the notion that oceanic carbon reservoirs, particularly the deep ocean, responded more slowly to atmospheric forcing.

In the latter part of MIS M2, a notable decline in $pCO_2$ is accompanied by an increase in $\delta^{13}C$ values for both benthic and planktic records. However, the magnitude of this recovery is greater in benthic foraminifera (**Fig. 6b**). The stronger amplitude in $\delta^{13}C_{BF}$ suggests a deep ocean response to increased ventilation and remineralised carbon drawdown, which may have lagged behind surface ocean reorganisation (Raymo et al., 1996). Meanwhile, $\delta^{13}C_{CO}$ values, while also increasing, exhibit a more modest recovery, consistent with a surface ocean that remained more sensitive to fluctuations in $pCO_2$ and biological productivity (Pagani et al., 2009). Following MIS M2, $\delta^{13}C_{BF}$ values remain relatively elevated throughout the mPWP (**Fig. 6b**). However, the persistently large $\delta^{13}C_{BF-CO}$ gradient (**Fig. 3a**) is consistent with model simulations showing reduced deep ocean overturning and widespread deep-ocean warming, indicating limited vertical mixing and poor deep-water ventilation and likely the accumulation of $^{12}C$-rich respired carbon in the deep ocean (Feng et al., 2020, Ford et al., 2025). In contrast, $\delta^{13}C_{CO}$ fluctuates more dynamically with $pCO_2$, implying that surface-ocean carbon cycling remained regionally variable and modulated by orbital-scale insolation changes (**Fig. 4i**), shifts in atmospheric circulation (e.g., wind-driven upwelling) (Beal et al., 2011), and variations in upper ocean stratification and nutrient delivery linked to ocean circulation patterns. This divergence between surface and deep-ocean carbon signals during the warm interval highlights the regional imprint of biogeochemical processes on globally integrated carbon reservoirs, consistent with evidence for reduced deep-water formation and ventilation in other ocean basins under mPWP conditions (Ford et al., 2022).

A parallel pattern is evident in the $\delta^{18}O$ records, where peaks in $pCO_2$ (**Fig. 6c**) consistently align with more negative $\delta^{18}O$ values in both benthic foraminifera (**Fig. 6e**) and coccoliths (**Fig. 6d**), indicating episodes of ocean warming. Given that $\delta^{18}O$ incorporates both temperature and global ice volume influences, the observed minima during $pCO_2$ peaks support interpretations of reduced global ice volume and increased SSTs during warm climate states (Lisiecki and Raymo, 2005). The synchronicity between $pCO_2$ maxima and $\delta^{18}O$ minima, especially in surface-dwelling coccoliths, underscores the sensitivity of upper ocean layers to rapid greenhouse gas forcing. These $\delta^{18}O_{CO}$ signals likely reflect immediate atmospheric–oceanic feedback, including enhanced stratification and reduced vertical mixing. Notably, in the modern western Indian Ocean, where Site U1476 is located, there is a near-equilibrium between the surface ocean and $pCO_2$, resulting in a minimal net air–sea $CO_2$ flux (Guo and Timmermans, 2024; Takahashi et al., 2009). This quasi-equilibrium condition highlights the

tight coupling between the ocean and atmosphere in the region. It illustrates how modern dynamics can inform the interpretation that $\delta^{18}O_{CO}$ variations at U1476 in the Pliocene were closely tied to greenhouse gas-driven changes in upper ocean structure and carbon exchange.

Similarly, $\delta^{18}O_{BF}$ values exhibit broadly synchronous trends that track $p$CO$_2$ variability (**Fig. 6e**), highlighting the strong climatic imprint of greenhouse forcing across the water column. However, $\delta^{18}O_{BF}$ values respond to combined eccentricity and obliquity pacing (**Fig. 4f**), consistent with the slower response times and thermal inertia of the deep ocean (Lisiecki and Raymo, 2005), whereas $\delta^{18}O_{CO}$ values are more strongly modulated by eccentricity (**Fig. 4l**), reflecting surface-layer sensitivity to insolation-driven changes in upper ocean stratification and temperature (**Fig. 5**). This further suggests that

while global SST patterns and ice volume were primarily paced by orbital forcing, regional responses, particularly in the Indian Ocean, were strongly influenced by feedbacks involving ocean dynamics, heat redistribution, and atmospheric circulation (Dowsett et al., 2016; Dowsett et al., 2013). Findings from this study therefore highlight the need for higher-resolution spatial and temporal integration of proxy data with model simulations, underscoring the importance of tropical basins in modulating global climate-carbon feedback mechanisms during past and potentially future warm climates.

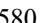

**Figure 6. Carbon and oxygen isotopes and atmospheric CO₂ ($p$CO₂) across the mid-Piacenzian Warm Period (mPWP) at Site U1476. (a) coccolith fraction δ¹³C, (b) benthic foraminifera δ¹³C, (c) $p$CO₂ (de la Vega et al., 2020), (d) coccolith fraction δ¹⁸O, (e) benthic foraminiferal δ¹⁸O. Glacial stages are shown by grey horizontal bars (Lisiecki and Raymo, 2005).**

## 4 Conclusions

This study presents the first high-resolution δ¹³C and δ¹⁸O records from coccoliths and benthic foraminifera, along with high-resolution coccolith assemblage data, and a low-resolution planktic foraminifera record spanning the Piacenzian at Site U1476. These combined records provide a vertically resolved reconstruction of upper-ocean stratification and carbon cycling in the western tropical Indian Ocean. Persistent vertical isotopic gradients indicate a thermally stratified upper ocean and an active, yet potentially nutrient-limited, biological pump. Distinct isotopic offsets between the *F. profunda*-dominated coccolithophore assemblage and *G. ruber* support photic zone niche partitioning and reveal the influence of deep-photic coccolithophores in biasing coccolith fraction isotopic signals, underscoring the need for taxon-specific reconstructions.

We demonstrate that high surface productivity during the mPWP did not imply efficient carbon export, as stratification likely inhibited nutrient resupply. The MIS M2 glacial event marks a transient collapse of stratification, highlighting the sensitivity of upper ocean structure and carbon sequestration to climate perturbations. Obliquity-paced variability in δ¹³C and δ¹⁸O records further links high-latitude orbital forcing to low-latitude stratification and export dynamics. The muted but coherent benthic response suggests deep-sea carbon cycling lagged behind surface processes, with implications for the timing of global carbon storage. By integrating these new tropical Indian Ocean records with the broader Piacenzian framework, this study provides a critical low-latitude perspective that complements well-documented Atlantic and Pacific reconstructions, advancing our understanding of ocean–carbon feedback in a warm, high-CO₂ world. These results establish the tropical Indian Ocean as an active node in global carbon cycling during the mPWP, providing new constraints on ocean–atmosphere feedback relevant to future high-$p$CO₂ scenarios.

## Data availability

All datasets generated in this study will be made publicly available via Zenodo upon publication. These include coccolith datasets comprising abundance, CaCO₃ content, mass, and stable isotope measurements (δ¹³C and δ¹⁸O) (DOIs: 10.5281/zenodo.16043100, 10.5281/zenodo.16043901, 10.5281/zenodo.16044409, 10.5281/zenodo.16044839), as well as planktic and benthic foraminifera stable isotope data (δ¹³C and δ¹⁸O) (DOIs: 10.5281/zenodo.16045236 and 10.5281/zenodo.16045489, respectively).

## Author contribution

DNT and IRH led the conceptualization of the study with input from LB and MAB. Formal analysis was conducted by DNT and LB. DNT acquired funding and administered the project with IRH. IRH provided supervision and, together with DNT, contributed resources for the study. Data visualization was done by DNT. DNT wrote the original draft, and all authors (DNT, IRH, LB, MAB, AN, and PRB) contributed to the review and editing of the manuscript.

## Competing interests

LB is a member of the Editorial Board of Climate of the Past. The authors declare that they have no other competing interests.

## Acknowledgments

This research used samples and data provided by IODP. We are thankful for the support from the scientific and technical crew of the R/V JOIDES Resolution IODP Expedition 361 and the IODP staff. This work is funded by the European Union's Horizon 2020 research and innovation programme under the Marie Sklodowska-Curie grant agreement No. 885498 awarded to DNT.

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
