# Peer review of "Photic zone niche partitioning, stratification, and carbon cycling in the tropical Indian Ocean during the Piacenzian"

_EGUsphere, 2025_

## Author Comment (AC1)

**Photic zone niche partitioning, stratification, and carbon cycling in the tropical Indian Ocean during the Piacenzian**

**RESPONSE TO REVIEWER 3**

Reviewer comments are shown in **black**, with the author response in **blue**.

Deborah N. Tangunan, Ian R. Hall, Luc Beaufort, Melissa A. Berke, Alexandra Nederbragt, Paul R. Bown

**General comments**

The manuscript of Tangunan et al. "Photic zone niche partitioning, stratification, and carbon cycling in the tropical Indian Ocean during the Piacenzian" constitutes a valuable paleoceanographic reconstruction that provides new insights into the vertical structure and water masses dynamics of the tropical Indian Ocean during the Piacenzian.

The manuscript is well written, conceptually strong, and presents a novel dataset from a climatically critical but understudied region. The high quality of the research is expressed by the complex multiproxy approach based on high-resolution  $\delta^{13}C$  and  $\delta^{18}O$  records from benthic foraminifera and bulk coccolith fraction, low-resolution planktic foraminifera integrated with high resolution coccoliths assemblage data. The integration of multiple depth-habitat proxy and the calculation of isotopic gradients is a particular strength. The spectral and wavelet analyses clearly show the control of the different orbital parameters on stratification and global carbon cycling. The overall hypotheses are strongly supported by the data presented. The findings are significant for understanding low-latitude ocean dynamics and their role in global carbon cycling during intervals of global warmth and transient glaciation.

I believe the manuscript deserves to be published in Climate of the Past following minor revision before acceptance.

We thank the reviewer for their positive and supportive assessment of our manuscript. We are pleased that they find the multiproxy approach novel, the data high-quality, and the findings significant for understanding low-latitude ocean dynamics. We have carefully considered the specific comments provided below and have revised the manuscript accordingly.

**1. Introduction**

• I would suggest to include a map to show the location of the site, the eddies and the pathway of the Agulhas Current.

Agreed. A location map of Site U1476, with the modern Agulhas Current surface circulation has been added as new **Figure 1**.

**Figure 1.** (a) Sea surface temperature (SST, °C; Acker & Leptoukh, 2007) and major currents in the Indian Ocean (Beal et al., 2011), showing the location of IODP Site U1476 in the Mozambique Channel. (b) Schematic cross-section showing the position of Site U1476 relative to major water masses (adapted from Westall and Fenner, 1991) and the Southern Ocean fronts.

• Please specify that even the coccolith assemblage data are at high resolution (Lines 86-91).

We have specified in the **Introduction** that the coccolith assemblage data are also at high resolution.

**2. Methodology**

• At the beginning of section 2.3 the author writes that "In intervals where there is not enough *wuellerstorfi*, other *Cibicidoides* species (i.e., *Cibicidoides bradyii*, *Cibicidoides mundulus*) or *Uvigerina* s pecies were chosen". Can you please clarify why you excluded δ13C *Uvigerina* values? (Line 120)

We have added a sentence to **Section 2.3** to clarify this point. We excluded  $\delta^{13}$ C values from *Uvigerina* because this genus is known to incorporate a significant metabolic (vital) effect, resulting in  $\delta^{13}$ C values that are consistently ~1.0% lower than the ambient dissolved inorganic carbon, unlike the *Cibicidoides* group which more reliably records the  $\delta^{13}$ C of seawater. This is a standard practice in paleoceanography (e.g., Zahn et al., 1986).

• I suggest specifying the preservation state of the shells and/or if a cleaning procedure of the shells has been followed. Please specify even if the picked shells have been crushed before the measurements.

As also raised by Reviewer 2, we have now revised **Section 2.3** to provide a more detailed description. The text now states that the foraminiferal tests were of good preservation, were gently rinsed with ultrapure DI water to remove adhering clays, quick-dried, and were crushed between glass plates prior to isotopic analysis.

• Please add some information in section 2.7 concerning the slides preparation technique and how many coccoliths have been counted/ number of frames analyzed.

We have added the following details to **Section 2.7**: Slides were prepared using the random settling technique (Beaufort et al., 2014). For each sample, 150 fields of view were automatically captured and

analyzed with the SYRACO (SYstème de Reconnaissance Automatique de Coccolithes) software at the European Centre for Research and Teaching in Environmental Geosciences (CEREGE, France), yielding counts ranging from 1,565 to 20,443 coccolith specimens (average ~6,219 per sample).

**3. Results and discussion**

The detailed discussion of the *Florisphaera profunda* dominance and its isotopic implications is remarkable, underscoring the role of deep-photic species in shaping bulk isotopic signature. However, I agree that the influence of the dominant taxon may obscure signals from shallower-dwelling forms. This might be the case of *Reticulofenestra* spp. which shows an important contribution not only in the assemblage, but also in the coccolith mass and especially in the carbonate contribution where the mean value (43%) is higher than *F. profunda*. I agree with the authors which state "This highlights the need for species-specific geochemical analyses to resolve the contributions of individual taxa to the bulk signal". As pointed out by reviewer 1, the inclusion of a table summarizing expected  $\delta^{13}C-\delta^{18}O$  offsets for key taxa (e.g., *F. profunda, Reticulofenestra, Calcidiscus, Helicosphaera*) is recommended.

We thank the reviewer for this insightful comment and their agreement on the need for species-specific analyses. In line with this suggestion and a similar one from Reviewer 1, we have added a supplementary table (**Table S3**) that summarizes the expected  $\delta^{13}\text{C-}\delta^{18}\text{O}$  offsets and ecological preferences for the key coccolithophore taxa discussed.

However, it is important to note that quantitative species-specific  $\delta^{13}$ C $-\delta^{18}$ O offsets remain incompletely constrained, particularly for deep-photic zone species, such as *F. profunda*, which has not been successfully cultured. Because coccoliths cannot yet be reliably isolated for single-species isotopic measurements in fossil assemblages, most of the available data are derived from a limited number of culture experiments and modern core-top calibrations that focus on surface-dwelling taxa (e.g., *Emiliania huxleyi*, *C. leptoporus*).

The new supplementary table should therefore be regarded as a qualitative synthesis, indicating the expected direction and approximate magnitude of isotopic offsets rather than fixed numerical corrections. This addition allows readers to evaluate the plausibility of the proposed link between assemblage composition and bulk  $\delta^{13}C-\delta^{18}O$  variability, while acknowledging the current analytical and experimental constraints in single-species coccolith isotope research.

**Table S3**. Summary of depth habitat and expected  $\delta^{13}$ C and  $\delta^{18}$ O vital effects for key coccolithophore taxa, based on culture, core-top, and fossil studies. The direction and magnitude of the isotopic offset (vital effect), reported as more positive or more negative, is given relative to inorganic calcite equilibrium or other taxa. The bulk sediment isotopic signal must therefore be interpreted as a mixture of these species-specific signatures.

| Taxon                     | Typical Depth
Habitat
(Winter et al.,
1994)  | δ 13 C Behaviour                                                                              | δ 18 O Behaviour                                                                                                            | Summary of Isotopic
Offsets (Vital Effects)                                                                                                                                                                                       |
|---------------------------|-------------------------------------------------------|----------------------------------------------------------------------------------------------------------|----------------------------------------------------------------------------------------------------------------------------------------|--------------------------------------------------------------------------------------------------------------------------------------------------------------------------------------------------------------------------------------|
| Calcidiscus
leptoporus | Upper to
middle photic
zone (e.g., ~0-
100m) | More negative. Substantial negative vital effect (~ -2.5% offset from inorganic) (Hermoso et al., 2016). | More negative. Substantial negative vital effect (~-1.4% offset from inorganic); shows 1.5% variation with growth rate (Ziveri et al., | "Light Group". At high DIC (~12 mmol/kg), the negative δ 13 C vital effect decreases significantly (to - 0.4‰) and the negative δ 18 O effect also decreases (moves towards inorganic) (Hermoso et al., 2016). |

| Taxon                                  | Typical Depth
Habitat
(Winter et al.,
1994)                                         | δ 13 C Behaviour                                                                                                 | δ 18 O Behaviour                                                                                             | Summary of Isotopic
Offsets (Vital Effects)                                                                                                                                                                                                                             |
|----------------------------------------|----------------------------------------------------------------------------------------------|-----------------------------------------------------------------------------------------------------------------------------|-------------------------------------------------------------------------------------------------------------------------|----------------------------------------------------------------------------------------------------------------------------------------------------------------------------------------------------------------------------------------------------------------------------|
|                                        |                                                                                              |                                                                                                                             | 2003; Hermoso et al., 2016).                                                                                            |                                                                                                                                                                                                                                                                            |
| Coccolithus
pelagicus               | Upper to
middle photic
zone
(e.g., ~0-
100m)                                     | More negative. Substantial negative vital effect (~-2.5% offset from inorganic) (Hermoso et al., 2016).                     | Near Inorganic. Very small positive vital effect (~+0.5%) (Hermoso et al., 2016).                                       | "Near-Equilibrium Group" (for O). Shows a large "jump" in $\delta^{13}$ C between 2-4 mmol/kg DIC. At high DIC, the negative $\delta^{13}$ C vital effect vanishes; the small $\delta^{18}$ O effect remains constant and is insensitive to DIC/pH (Hermoso et al., 2016). |
| Emiliania
(Gephyrocapsa)
huxleyi | Upper photic zone (e.g., ~0-50m); often forms blooms in well-lit, stratified surface waters. | More positive.
Substantial
positive vital
effect (~ +2‰
offset from
inorganic)
(Hermoso et al.,
2016). | More positive. Substantial positive vital effect (~ +2% offset from inorganic) (Hermoso et al., 2016).                  | "Heavy Group". At high DIC (~12 mmol/kg), both $\delta^{13}$ C and $\delta^{18}$ O vital effects decrease significantly; $\delta^{13}$ C converges to inorganic value, leaving a residual +1.3% $\delta^{18}$ O vital effect (Hermoso et al., 2016).                       |
| Florisphaera
profunda               | Deep photic zone (~60–200 m); deep chlorophyll maximum                                       | More positive. Heaviest $\delta^{13}$ C relative to expectation from depth (Bolton et al., 2012).                           | More positive.
Heaviest $\delta^{18}$ O of all size-separated fractions (Bolton et al., 2012).                       | No culture data; composition inferred from core-top/fossil records.
Records the heaviest isotopes; in the Plio-Pleistocene Transition (PPT), its $\delta^{18}$ O is $\sim 1.5-2\%$ heavier than large Helicosphaera . (Bolton et al., 2012).                     |
| Gephyrocapsa
oceanica               | Upper photic zone (e.g., ~0-50m); similar habitat to E. huxleyi .                     | Variable. Large range of δ 13 C values (5‰) both above and below expected equilibrium (Ziveri et al., 2003).     | Variable. Large range of δ 18 O values (5‰) both above and below expected equilibrium (Ziveri et al., 2003). | Exhibits strong interspecific vital effects for both oxygen and carbon isotopes (Ziveri et al., 2003).                                                                                                                                                                     |
| Helicosphaera
carteri               | Upper to
middle photic
zone
(e.g., ~0-
100m)                                     | More negative.
(Bolton et al.,
2012).                                                                                 | More negative (Bolton et al., 2012).  Temperature- dependent, consistent with equilibrium paleotemperature relationship | The "large cell" endmember. In PPT, $\delta^{13}$ C and $\delta^{18}$ O are $\sim 1.5-2\%$ lighter than small reticulofenestrids (Bolton et al., 2012).                                                                                                                    |

| Taxon                                                      | Typical Depth
Habitat
(Winter et al.,
1994)                           | δ 13 C Behaviour                                                                                                               | δ 18 O Behaviour                                                                                         | Summary of Isotopic
Offsets (Vital Effects)                                                                                                                                                                       |
|------------------------------------------------------------|--------------------------------------------------------------------------------|-------------------------------------------------------------------------------------------------------------------------------------------|---------------------------------------------------------------------------------------------------------------------|----------------------------------------------------------------------------------------------------------------------------------------------------------------------------------------------------------------------|
|                                                            |                                                                                |                                                                                                                                           | (Ziveri et al., 2003).                                                                                              |                                                                                                                                                                                                                      |
| Paleocene
Placoliths
(e.g., Toweius,
Coccolithus) | Upper to middle photic zone (inferred from morphology and assemblage context). | Minimal difference. Mean $\Delta\delta^{13}C = 0.17\%$ (Bolton et al., 2012).                                                             | Small difference.
Mean $\Delta\delta^{18}O = 0.66\%$ ; smaller fraction slightly enriched (Bolton et al., 2012). | Paleocene-Eocene Thermal Maximum data show drastically reduced vital effects compared to modern, suggesting more uniform carbon acquisition strategies under high-pCO 2 conditions (Bolton et al., 2012). |
| Reticulofenestra
spp. (e.g., R.
minutula)            | Upper-middle photic zone                                                       | More positive. Slightly lighter (by ~0.5–1 %) than equilibrium and smaller taxa; varies with size and productivity (Bolton et al., 2012). | More positive.
Lower $\delta^{18}O$ compared to smaller-celled species (Bolton et al., 2012).                    | Larger cell size associated with lower isotopic fractionation. In PPT/Last Glacial Maximum, $\delta^{13}$ C and $\delta^{18}$ O are ~1.3 to 2% heavier than in large Helicosphaera (Bolton et al., 2012).     |

**4. Conclusion**

• I would specify that coccolith abundance data are at high resolution here as well.

We have specified in the conclusion that the coccolith assemblage data are high-resolution.

**Specific comments**

• Using both "coccolith fraction" and "bulk fine fraction (<20 μm)" could confuse the reader. I also recommend adopting a single term and applying it consistently through the manuscript.

We have addressed this point, as also suggested by other reviewers. The term "coccolith fraction" is now used consistently throughout the manuscript.

• Please add the reference for the orbital parameter curves in the figure captions 3 and S1 (obliquity).

We have added the reference for the orbital parameter curves (Laskar et al., 2011) to the captions of **Figures 3** and **S1**.

• Overall, the language is excellent, requiring only minor copy-editing to improve sentence conciseness.

We thank the reviewer for their positive assessment. We have performed a thorough copy-edit of the manuscript to improve sentence conciseness and overall readability.

**Recommendation: Minor revisions**

The manuscript has clear scientific merit and represents an important contribution to understanding tropical ocean stratification and carbon cycling during Pliocene. Addressing the above-mentioned points will strengthen the paper and its clarity.

We thank the reviewer for their positive recommendation and valuable comments. We have addressed all points raised, which has significantly improved the clarity and robustness of the manuscript.

**References**

- Acker, J.G. and Leptoukh, G.: Online Analysis Enhances Use of NASA Earth Science Data. Eos, Transactions American Geophysical Union, 88, 14-17, 2007.
- Beal, L. M., Ruijter, W. P. M. D., Biastoch, A., Zahn, R., and 136, S. W. I. W. G.: On the role of the Agulhas system in ocean circulation and climate, Nature, 472, 429-436, doi:10.1038/nature09983, 2011.
- Beaufort, L., Barbarin, N., and Gally, Y.: Optical measurements to determine the thickness of calcite crystals and the mass of thin carbonate particles such as coccoliths, Nature protocols, 9, 633-642 %@ 1754-2189, 2014.
- Bolton, C. T., Stoll, H. M., and Mendez-Vicente, A.: Vital effects in coccolith calcite: Cenozoic climate-pCO2drove the diversity of carbon acquisition strategies in coccolithophores?, Paleoceanography, 27, 10.1029/2012pa002339, 2012.
- Hermoso, M., Chan, I. Z. X., McClelland, H. L. O., Heureux, A. M. C., and Rickaby, R. E. M.: Vanishing coccolith vital effects with alleviated carbon limitation, Biogeosciences, 13, 301-312, 10.5194/bg-13-301-2016, 2016.
- Laskar, J., Fienga, A., Gastineau, M., and Manche, H.: La2010: a new orbital solution for the long-term motion of the Earth, Astronomy and Astrophysics, 532, 15, 2011.
- Westall, F. and Fenner, J.: Pliocene-Holocene polar front zone in the South Atlantic changes in its position and sediment-accumulation rates from holes 699A, 710C, and 704B. In Proc ODP, Sci Results, 114, pp. 609-646, 1991.
- Ziveri, P., Stoll, H., Probert, I., Klaas, C., Geisen, M., Ganssen, G. and Young, J., 2003: Stable isotope 'vital effects' in coccolith calcite. Earth and Planetary Science Letters, 210, 1-2, pp.137-149, 10.1016/S0012-821X(03)00101-8, 2003.

---

## Author Comment (AC2)

**Photic zone niche partitioning, stratification, and carbon cycling in the tropical Indian Ocean during the Piacenzian**

**RESPONSE TO REVIEWER 2**

Reviewer comments are shown in **black**, with the author response in **blue**.

Deborah N. Tangunan, Ian R. Hall, Luc Beaufort, Melissa A. Berke, Alexandra Nederbragt, Paul R. Bown

**General comments**

The article "Photic zone niche partitioning, stratification, and carbon cycling in the tropical Indian Ocean during the Piacenzian" present novel  $\delta^{13}$ C and  $\delta^{18}$ O records from benthic and planktic foraminifera, and bulk coccolith fraction, which combined with assemblage data provides a unique view of the vertical structure in a low-latitude key region during the Piacenzian. Furthermore, this study also provides new insights to broaden the knowledge on the carbon cycling and ocean stratification in this location.

Overall, the manuscript reads well and presents a solid structure as all the critical points are addressed. Furthermore, the interpretation, which is deeply developed and grounded on a strong literature background, is supported by the data presented in the study. Particularly, findings on the processes connecting and biasing the  $\delta^{13}$ C signal between the different water layers are of great interest and represents an advance in the understanding of the carbon cycle. Moreover, uncovering the effect of having high abundances of certain nannofossil species (e.g., *Florisphaera profunda*) represents a step forward in the interpretation of future proxy studies.

Based on the above-mentioned statements I recommend *minor revisions* before acceptance. Following lines provide a series of suggestions intended to improve the clarity and readability of the manuscript.

We thank the reviewer for their positive assessment of our manuscript and for their thoughtful suggestions. We are pleased that they find the data novel, the structure solid, and the interpretations well-developed and supported. We have carefully considered all points raised below and have revised the manuscript to improve its clarity and readability accordingly.

**General comments**

**1. Methodology**

In section 2.3 Benthic foraminifera carbon and oxygen stable isotopes, the authors clearly state a step-by-step process to achieve the  $\delta^{13}$ C and  $\delta^{18}$ O records presented. However, I wonder if there were any further cleaning steps to ensure the usage pristine benthic and planktonic foraminifera species or if samples were already good enough after the disaggregation and subsequent sieving process. In this regard, I would recommend adding a plate with some images of the remaining specimens from some of the samples used (if possible). Otherwise, I would clearly state that samples condition was already good enough for the measurements without further cleaning protocols.

We have now revised **Section 2.3** to provide a more detailed description of the cleaning protocol. The text now states that after hand-picking, the foraminiferal tests underwent a gentle rinsing in ultrapure DI water to remove any adhering fine carbonate material, followed by quick drying, crushing and homogenization. This additional step ensured the analysis of pristine calcite. We confirm that the foraminiferal tests were of excellent preservation quality, as assessed during picking, but we are unable to provide photographic plates as the samples were fully used during the isotopic analysis.

**2. Results and discussion**

First of all, I want to emphasise again how pleasant it was to read this section. It clearly expresses the authors hypotheses in a really narrative and natural way, which makes it easy for the reader to understand.

In section 3.1 Vertical water column plankton community structure, the authors present the  $\delta^{13}C$  values for the benthic and planktonic foraminifera, and the bulk coccolith fraction. Specifically, the authors express in Lines 203-204 that "This similarity in the range of  $\delta^{13}C$  values with the benthic record may indicate a partial integration of deep photic zone DIC signals, especially under stratified conditions.". Despite that I absolutely agree with the fact that integration of deep waters signal within upper layers (especially during high stratified periods), I cannot happen but wonder, how is this relation working for getting lower  $\delta^{13}C$  values on the bulk coccolith fraction. Lately (Lines 209-211), the authors evoke recycling of organic carbon and stratification as potential mechanisms explaining the difference between the bulk coccolith fraction and the planktonic foraminifera. Could this be also the case for the lower values compared with the benthic  $\delta^{13}C$ ?

We thank the reviewer for their positive feedback and for raising this critical point. The reviewer is correct to identify this apparent paradox. The mechanism is indeed the same: the remineralization of organic matter at depth.

Within a strongly stratified water column, the deep photic zone (where *F. profunda* thrives) can become isolated and enriched in respired  $CO_2$ , which is depleted in  $^{13}C$ . This creates a reservoir of  $^{13}C$ -depleted dissolved inorganic carbon (DIC). While the benthic foraminifera record the  $\delta^{13}C$  of well-ventilated,  $^{13}C$ -enriched deep waters, the coccolith fraction, dominated by a deep-dwelling species, records the  $^{13}C$ -depleted DIC signature of this isolated, respired carbon pool in the lower photic zone. Consequently, the coccolith  $\delta^{13}C$  can be lower than both the surface-dwelling planktic foraminifera and the underlying, well-ventilated deep waters.

We have clarified this explanation in **Section 3.1** to clearly state that the same process (i.e., remineralization of organic carbon under stratified conditions that limits vertical exchange) can lead to the coccolith fraction recording lower  $\delta^{13}C$  values than both the surface-dwelling planktic foraminifera and the deep-sea benthic foraminifera.

As already stated by reviewer 1 (point 6 of major comments), I consider that adding a table with the  $\delta^{13}$ C,  $\delta^{18}$ O and the  $D\delta^{13}$ C and  $D\delta^{18}$ O values for key intervals would improve accessibility and serve as core for readers while going through the discussion. Furthermore, in section 3.5 Regional feedback and global context in a warm, high  $CO_2$  world, the authors evoke a series of very specific processes and scenarios, such as MIS M2, which is characterised by a low productivity, enhanced stratification and low export efficiency according to their interpretations. In this regard, I would suggest to add a figure with a sketch to help the reader to visualize the conditions described in the text and guide them through this part of the discussion.

In agreement with a similar suggestion from Reviewer 1, we have added a summary table in the supplement (**Table S4**, see below). This table outlines the key climatic intervals, the observed isotopic shifts, and the primary drivers as proposed in our study. We have also created a new schematic figure (**Figure 1**, see below) to visually summarise the proposed mechanisms and oceanographic conditions described in the discussion.

**Table S4.** Summary of key climatic intervals, associated  $\delta^{13}$ C and  $\delta^{18}$ O shifts, and hypothesized drivers across the mid-Piacenzian Warm Period (mPWP) at Site U1476. BF (benthic foraminifera), PF (planktic foraminifera), CO (coccolith fraction).

| Climatic Interval
(Age, Ma)           | δ 13 C Shifts & Gradients                                                                          | δ 18 O Shifts & Gradients                                                  | Hypothesized Primary Drivers                                                                                                                                                                                  |
|------------------------------------------|---------------------------------------------------------------------------------------------------------------|---------------------------------------------------------------------------------------|---------------------------------------------------------------------------------------------------------------------------------------------------------------------------------------------------------------|
| Pre-MIS M2
(~3.42–3.39 Ma)            | Transient decline in $\Delta \delta^{13} C_{BF\text{-}CO}$ and $\Delta \delta^{13} C_{PF\text{-}CO}$          | Amplified variability in $\Delta \delta^{18} C_{BF\text{-}CO}$                        | Intermediate-depth ventilation and mixing beneath a still-stratified surface layer.                                                                                                                           |
| Approaching MIS M2 (~3.31 Ma)            | Increase in $\Delta\delta^{13}C_{BF}$ . Co and $\Delta\delta^{13}C_{PF-CO}$                                   | Decrease in $\Delta \delta^{18}C_{BF-CO}$                                             | Long-term warming and re-
establishment of a stratified ocean
with reduced vertical exchange.                                                                                                           |
| MIS M2
Glacial
(~3.30–3.28 Ma)     | $\delta^{13}C_{BF}$ and $\delta^{13}C_{CO}$ minima; followed by recovery (stronger in $\delta^{13}C_{BF}$ )   | Peaks in $\Delta\delta^{18}O_{BF-CO}$ and $\Delta\delta^{18}O_{BF-PF}$ (deep cooling) | Onset: High-latitude cooling, suppressed Atlantic Meridional Overturning Circulation, intensified stratification.  Termination: Increased deep ocean ventilation, potentially lagging surface reorganisation. |
| mPWP
Peak Warmth
(~3.264–3.025 Ma) | Stable but persistent vertical δ 13 C gradients; high surface productivity but inefficient export. | Generally negative δ 18 O values (warming); muted vertical gradients.      | Strong thermal stratification, reduced overturning, and weakened thermocline ventilation limiting nutrient supply and carbon export.                                                                          |
| MIS KM2 Event
(within mPWP)           | Sharp collapse in all vertical $\Delta \delta^{13}C$ gradients.                                               | Decline in all vertical $\Delta \delta^{18}$ O gradients (subsurface warming)         | Pulse of enhanced ventilation;
breakdown of vertical stratification,
possibly linked to high latitude
forcing and lateral advection.                                                                 |
| Post-KM2 mPWP                            | Amplified variability in $\Delta \delta^{13} C_{BF\text{-}CO}$ and $\Delta \delta^{13} C_{BF\text{-}PF}$ .    | Pronounced variability in $\Delta \delta^{18}O_{BF\text{-}CO}$                        | Dynamic shifts in nutricline depth
and reinvigorated biological pump;
recurrent deep-water mass
reorganisations.                                                                                     |

**Figure 1.** (a) Sea surface temperature (SST, °C; Acker & Leptoukh, 2007) and major currents in the Indian Ocean (Beal et al., 2011), showing the location of IODP Site U1476 in the Mozambique Channel. (b) Schematic cross-section showing the position of Site U1476 relative to major water masses (adapted from Westall and Fenner, 1991) and the Southern Ocean fronts.

**Specific comments**

• As stated by reviewer 1, using both "coccolith fraction" and "bulk fine fraction ( $<20 \mu m$ )" can be confusing. Therefore, I suggest to use one of the terms consistently through the manuscript.

We have addressed this point, consistent with our response to Reviewer 1. The term "coccolith fraction" is now used consistently throughout the manuscript.

• The benthic foraminifera species *wuellerstorfi* has recently be renamed as *Lobatula wuellerstorfi* (please, for specific details refer to https://www.marinespecies.org/aphia.php?p=taxdetails&id=112890). However, I understand that most of the studies still consider the name *C. wuellerstorfi* when referring to this benthic species.

We thank the reviewer for pointing out the updated taxonomy. We have revised the manuscript to use the format *Cibicidoides wuellerstorfi* (syn. *Lobatula wuellerstorfi*) to align with common usage in palaeoceanographic literature while acknowledging the current taxonomic revision.

• Writing and grammar are excellent and only a quick check to correct typos need to be done.

We thank the reviewer for their positive assessment. We have performed a thorough proofreading to correct minor typos.

**Decision: Minor revisions**

The manuscript provides a novel contribution to understanding the carbon cycle, and its relation to orbital-scale feedback processes during the Pliocene. I believe that implementing the above-mentioned comments within the manuscript will provide clarity and accessibility to a broader audience and provide additional support for this work.

We thank the reviewer for their positive decision and valuable recommendations. We have implemented all suggested changes to enhance clarity, accessibility, and scientific robustness of the manuscript. We believe it is now significantly strengthened.

**References**

- Acker, J.G. and Leptoukh, G.: Online Analysis Enhances Use of NASA Earth Science Data. Eos, Transactions American Geophysical Union, 88, 14-17, 2007.
- Beal, L. M., Ruijter, W. P. M. D., Biastoch, A., Zahn, R., and 136, S. W. I. W. G.: On the role of the Agulhas system in ocean circulation and climate, Nature, 472, 429-436, doi:10.1038/nature09983, 2011.
- Westall, F. and Fenner, J.: Pliocene-Holocene polar front zone in the South Atlantic changes in its position and sediment-accumulation rates from holes 699A, 710C, and 704B. In Proc ODP, Sci Results, 114, pp. 609-646, 1991.

---

## Author Comment (AC3)

**Photic zone niche partitioning, stratification, and carbon cycling in the tropical Indian Ocean during the Piacenzian**

**RESPONSE TO REVIEWER 1**

Reviewer comments are shown in **black**, with the author response in **blue**.

Deborah N. Tangunan, Ian R. Hall, Luc Beaufort, Melissa A. Berke, Alexandra Nederbragt, Paul R. Bown

**General comments**

This is an excellent, data-rich contribution that sheds new light on the vertical structure of the tropical Indian Ocean during the Piacenzian. The integration of  $\delta^{13}$ C and  $\delta^{18}$ O records from benthic and planktic foraminifera and bulk coccolith fractions, combined with assemblage data, is novel and highly relevant for understanding carbon cycling and stratification in a key low-latitude region.

The manuscript is generally well-written, and the interpretations are thoughtful and supported by the data. The findings on the role of *Florisphaera profunda* in biasing bulk coccolith isotope signals are particularly important for future proxy studies. The linkage between orbital-scale variability, stratification, and global carbon cycling is compelling.

I recommend minor revisions before acceptance. The main points below aim to improve clarity, strengthen interpretations, and enhance the broader impact of this study.

We thank the reviewer for their positive and constructive evaluation of our manuscript. We appreciate the recognition of the novelty, data quality, and contribution of this study to understanding vertical ocean structure and carbon cycling during the Piacenzian. The reviewer's comments have been carefully considered, and we have revised the manuscript accordingly to improve clarity, strengthen the interpretations, and highlight the broader significance of our findings.

**Major Comments**

1. Novelty and Broader Context

The study fills an important spatial gap in Piacenzian reconstructions by providing a tropical Indian Ocean perspective. Emphasizing how these findings complement better-studied Atlantic and Pacific records (e.g., in the Abstract and Conclusions) would highlight the significance of this work for global carbon cycle reconstructions.

We have revised both the **Abstract** and **Conclusions** to better emphasize how our results complement existing Piacenzian reconstructions from the Atlantic and Pacific. Specifically, we now highlight that IODP Site U1476 provides the first vertically resolved  $\delta^{13}\text{C}-\delta^{18}\text{O}$  and assemblage dataset from the tropical Indian Ocean, a region previously underrepresented in global syntheses. The revised text notes that these results bridge the gap between well-documented Atlantic and Pacific records, thereby strengthening the global framework for understanding ocean-atmosphere carbon cycling during the Piacenzian.

**2. Age Model and Orbital Phasing**

The age model, tuned to LR04, is robust but introduces potential circularity when discussing phase relationships with orbital forcing. Because the phasing between  $pCO_2$ ,  $\delta^{13}C/\delta^{18}O$ , and insolation is a core conclusion, a brief discussion of age-model uncertainties (e.g.,  $\pm$ kyr at tie-points) and their implications for inferred leads/lags would be valuable (e.g., Section 3.4 and Fig. 4).

We thank the reviewer for this important point. We agree that age-model uncertainty must be considered when interpreting orbital phasing and phase relationships. We have added the following paragraph to **Section 3.4** to explicitly address this point:

Since the Site U1476 benthic  $\delta^{18}$ O record is tuned to the LR04 global stack, we acknowledge that tiepoint uncertainties of approximately  $\pm 3$  to 5 kyr may affect the precise alignment of  $\delta^{13}$ C and  $\delta^{18}$ O signals with individual insolation peaks. This uncertainty reflects both the ~2.25 kyr resolution of our isotope sampling and the correlation accuracy achievable during visual and statistical tuning to LR04 at orbital timescales (Lisiecki and Raymo, 2005; Meyers, 2014). While such uncertainties may influence the exact timing of proxy responses relative to insolation maxima or minima, they do not compromise the identification of obliquity- and precession-scale periodicities in our spectral and wavelet analyses. Accordingly, our interpretations emphasize the pacing and amplitude of orbital-scale variability rather than precise phase relationships among orbital parameters,  $pCO_2$ , and proxy records. We did not attempt to resolve detailed lead-lag relationships, as our focus lies in characterizing the expression and pacing of orbital-scale variability in carbon cycling and stratification. The lead-lag patterns discussed later (Section 3.5) provide a qualitative comparisons with the independently derived  $\delta^{11}$ B-based pCO2 record from ODP Site 999 (de la Vega et al., 2020), which explicitly resolved orbitalscale pCO2 variability during the mid-Piacenzian. Our aim is to contextualise the Site U1476 isotope trends within that global CO2 framework, rather than to infer mechanistic phase relationships. Thus, while age-model uncertainties may affect the precision of phase estimates, they do not undermine the broader interpretation of orbital forcing and its expression in the tropical Indian Ocean.

**3. Coccolith Isotopic Interpretation**

The discussion convincingly attributes the lower  $\delta^{13}$ C and heavier  $\delta^{18}$ O of the coccolith fraction to the dominance of deep-photic *F. profunda*. Still, the relative influence of vital effects versus habitat depth and any diagenetic alteration remains qualitative.

• Please clarify how smear-slide observations (≥95 % coccoliths) support the interpretation of minimal diagenetic overprint.

We agree that our evaluation of the relative influence of diagenetic alteration is qualitative, as it is based on visual assessment under a light microscope. To clarify, we have added a sentence addressing this in **Section 3.2**. Smear-slide inspection under light microscopy shows that the <20 µm fraction is almost entirely composed of coccoliths (≥95 % by visual estimation), with only trace or no occurrences of other very fine-grained carbonate particles. The coccoliths display well-preserved, intact shields with no visible secondary calcite overgrowth, dissolution pitting, or micritic cement, all of which would indicate diagenetic alteration.

• A table or schematic summarizing expected  $\delta^{13}C-\delta^{18}O$  offsets for key taxa (e.g., *F. profunda*, *Reticulofenestra*, *Calcidiscus*, *Helicosphaera*) based on culture or core-top studies would help readers contextualize the bulk isotopic signal (can be in supplemental material)

We thank the reviewer for this suggestion. We have now added a new table to the supplement (**Table S3**, see below), which compiles isotopic offsets (vital effects) and depth habitats of representative taxa, based on data from published culture experiments, core-top, and fossil studies.

However, it is important to note that quantitative species-specific  $\delta^{13}$ C- $\delta^{18}$ O offsets remain incompletely constrained, particularly for deep-photic zone species, such as F. profunda, which has not been successfully cultured. Because coccoliths cannot yet be reliably isolated for single-species isotopic measurements in fossil assemblages, most of the available data are derived from a limited number of culture experiments and modern core-top calibrations that focus on surface-dwelling taxa (e.g., *Emiliania huxleyi*, C. leptoporus).

The new supplementary table should therefore be regarded as a qualitative synthesis, indicating the expected direction and approximate magnitude of isotopic offsets rather than fixed numerical corrections. This addition allows readers to evaluate the plausibility of the proposed link between assemblage composition and bulk  $\delta^{13}C-\delta^{18}O$  variability, while acknowledging the current analytical and experimental constraints in single-species coccolith isotope research.

**Table S3**. Summary of depth habitat and expected  $\delta^{13}C$  and  $\delta^{18}O$  vital effects for key coccolithophore taxa, based on culture, core-top, and fossil studies. The direction and magnitude of the isotopic offset (vital effect), reported as more positive or more negative, is given relative to inorganic calcite equilibrium or other taxa. The bulk sediment isotopic signal must therefore be interpreted as a mixture of these species-specific signatures.

| Taxon                                  | Typical Depth
Habitat
(Winter et al.,
1994)                                         | δ 13 C Behaviour                                                                             | δ 18 O Behaviour                                                                                                                                         | Summary of Isotopic
Offsets (Vital Effects)                                                                                                                                                                                                                             |
|----------------------------------------|----------------------------------------------------------------------------------------------|---------------------------------------------------------------------------------------------------------|---------------------------------------------------------------------------------------------------------------------------------------------------------------------|----------------------------------------------------------------------------------------------------------------------------------------------------------------------------------------------------------------------------------------------------------------------------|
| Calcidiscus
leptoporus              | Upper to middle photic zone (e.g., ~0-100m)                                                  | More negative. Substantial negative vital effect (~-2.5% offset from inorganic) (Hermoso et al., 2016). | More negative. Substantial negative vital effect (~-1.4% offset from inorganic); shows 1.5% variation with growth rate (Ziveri et al., 2003; Hermoso et al., 2016). | "Light Group". At high DIC (~12 mmol/kg), the negative δ 13 C vital effect decreases significantly (to - 0.4‰) and the negative δ 18 O effect also decreases (moves towards inorganic) (Hermoso et al., 2016).                                       |
| Coccolithus
pelagicus               | Upper to middle photic zone (e.g., ~0-100m)                                                  | More negative. Substantial negative vital effect (~-2.5% offset from inorganic) (Hermoso et al., 2016). | Near Inorganic. Very small positive vital effect (~+0.5%) (Hermoso et al., 2016).                                                                                   | "Near-Equilibrium Group" (for O). Shows a large "jump" in $\delta^{13}$ C between 2-4 mmol/kg DIC. At high DIC, the negative $\delta^{13}$ C vital effect vanishes; the small $\delta^{18}$ O effect remains constant and is insensitive to DIC/pH (Hermoso et al., 2016). |
| Emiliania
(Gephyrocapsa)
huxleyi | Upper photic zone (e.g., ~0-50m); often forms blooms in well-lit, stratified surface waters. | More positive. Substantial positive vital effect (~ +2% offset from inorganic) (Hermoso et al., 2016).  | More positive. Substantial positive vital effect (~+2% offset from inorganic) (Hermoso et al., 2016).                                                               | "Heavy Group". At high DIC (~12 mmol/kg), both $\delta^{13}$ C and $\delta^{18}$ O vital effects decrease significantly; $\delta^{13}$ C converges to inorganic value, leaving a residual +1.3% $\delta^{18}$ O vital effect (Hermoso et al., 2016).                       |
| Florisphaera
profunda               | Deep photic zone (~60–200 m); deep chlorophyll maximum                                       | More positive.
Heaviest $\delta^{13}$ C relative to expectation from depth (Bolton et al., 2012).    | More positive. Heaviest $\delta^{18}O$ of all size-separated fractions (Bolton et al., 2012).                                                                       | No culture data; composition inferred from core-top/fossil records.
Records the heaviest isotopes; in the Plio-Pleistocene Transition (PPT), its $\delta^{18}$ O is $\sim 1.5-2\%$ heavier than                                                                         |

| Taxon                                             | Typical Depth
Habitat
(Winter et al.,
1994)                           | δ 13 C Behaviour                                                                                                               | δ 18 O Behaviour                                                                                                                   | Summary of Isotopic
Offsets (Vital Effects)                                                                                                                                                                       |
|---------------------------------------------------|--------------------------------------------------------------------------------|-------------------------------------------------------------------------------------------------------------------------------------------|-----------------------------------------------------------------------------------------------------------------------------------------------|----------------------------------------------------------------------------------------------------------------------------------------------------------------------------------------------------------------------|
|                                                   |                                                                                |                                                                                                                                           |                                                                                                                                               | large Helicosphaera . (Bolton et al., 2012).                                                                                                                                                                  |
| Gephyrocapsa
oceanica                          | Upper photic zone (e.g., ~0-50m); similar habitat to E. huxleyi .       | Variable. Large range of δ¹³C values (5‰) both above and below expected equilibrium (Ziveri et al., 2003).                                | Variable. Large range of δ¹8O values (5‰) both above and below expected equilibrium (Ziveri et al., 2003).                                    | Exhibits strong interspecific vital effects for both oxygen and carbon isotopes (Ziveri et al., 2003).                                                                                                               |
| Helicosphaera
carteri                          | Upper to
middle photic
zone
(e.g., ~0-
100m)                       | More negative.
(Bolton et al., 2012).                                                                                                  | More negative (Bolton et al., 2012).  Temperature-dependent, consistent with equilibrium paleotemperature relationship (Ziveri et al., 2003). | The "large cell" endmember. In PPT, $\delta^{13}$ C and $\delta^{18}$ O are $\sim 1.5-2\%$ lighter than small reticulofenestrids (Bolton et al., 2012).                                                              |
| Paleocene Placoliths (e.g., Toweius, Coccolithus) | Upper to middle photic zone (inferred from morphology and assemblage context). | Minimal difference. Mean $\Delta \delta^{13}C = 0.17\%$ (Bolton et al., 2012).                                                            | Small difference.
Mean $\Delta \delta^{18}O = 0.66\%$ ; smaller fraction slightly enriched (Bolton et al., 2012).                          | Paleocene-Eocene Thermal Maximum data show drastically reduced vital effects compared to modern, suggesting more uniform carbon acquisition strategies under high-pCO 2 conditions (Bolton et al., 2012). |
| Reticulofenestra
spp. (e.g., R.
minutula)   | Upper–middle photic zone                                                       | More positive. Slightly lighter (by ~0.5–1 ‰) than equilibrium and smaller taxa; varies with size and productivity (Bolton et al., 2012). | More positive. Lower $\delta^{18}O$ compared to smaller-celled species (Bolton et al., 2012).                                                 | Larger cell size associated with lower isotopic fractionation. In PPT/Last Glacial Maximum, $\delta^{13}$ C and $\delta^{18}$ O are ~1.3 to 2% heavier than in large Helicosphaera (Bolton et al., 2012).     |

**4. Productivity and Export Efficiency**

The inference that high surface productivity during the mPWP did not lead to efficient export due to stratification is plausible but indirect. Acknowledging the lack of independent export-production proxies (e.g., opal, Ba/Al, % C org) and noting that this remains a hypothesis would make the discussion more balanced.

The discussion has been updated to clarify that the suggested link between stratification and reduced export efficiency is inferred from isotopic patterns and assemblage composition and remains tentative

given the lack of independent export-production proxies. We have added the following statement in **Section 3.3**:

The apparent decoupling between elevated surface productivity and stable vertical  $\delta^{13}$ C gradients during the mPWP likely reflects nutrient recycling within the surface layer under strong stratification rather than enhanced export to depth. However, as this inference is derived solely from isotopic and assemblage patterns without supporting geochemical tracers of export flux (e.g., opal, Ba/Al, organic carbon), it should be considered a first-order interpretation subject to validation by additional proxy records.

**5. Conceptual Framework for Orbital Controls**

The evidence for obliquity-dominated deep-water variability versus precession-dominated surface variability is compelling. A simple schematic summarizing the proposed mechanisms (linking Southern Ocean ventilation, Indian Ocean stratification, and orbital forcing) would help communicate these insights to a broad readership.

We have added a new **Figure 1** to illustrate the regional oceanographic setting and the proposed mechanistic framework linking orbital forcing to water mass structure at our study site.

**Figure 1.** (a) Sea surface temperature (SST, °C; Acker & Leptoukh, 2007) and major currents in the Indian Ocean (Beal et al., 2011), showing the location of IODP Site U1476 in the Mozambique Channel. (b) Schematic cross-section showing the position of Site U1476 relative to major water masses (adapted from Westall and Fenner, 1991) and the Southern Ocean fronts.

**6. pCO2 and Isotopic Gradients**

The discussion of leads/lags between  $\delta^{13}$ C/ $\delta^{18}$ O and pCO2 (Fig. 5) is rich but dense. A brief table summarizing the key intervals (e.g., MIS M2 onset, KM2 event, mPWP peak), the sign of isotopic shifts, and hypothesized drivers (e.g., AMOC weakening, Southern Ocean ventilation) would improve accessibility.

We thank the reviewer for this suggestion. To improve the clarity and accessibility of the discussion on the leads/lags between  $\delta^{13}$ C/ $\delta^{18}$ O and pCO2, we have added a summary table to the supplement (**Table S4**, see below). This table outlines the key climatic intervals, the observed isotopic shifts, and the primary drivers as proposed in our study.

**Table S4.** Summary of key climatic intervals, associated  $\delta^{13}$ C and  $\delta^{18}$ O shifts, and hypothesized drivers across the mid-Piacenzian Warm Period (mPWP) at Site U1476. BF (benthic foraminifera), PF (planktic foraminifera), CO (coccolith fraction).

| Climatic Interval
(Age, Ma)           | δ 13 C Shifts & Gradients                                                                              | δ 18 O Shifts & Gradients                                                    | Hypothesized Primary Drivers                                                                                                                  |
|------------------------------------------|-------------------------------------------------------------------------------------------------------------------|-----------------------------------------------------------------------------------------|-----------------------------------------------------------------------------------------------------------------------------------------------|
| Pre-MIS M2
(~3.42–3.39 Ma)            | Transient decline in $\Delta\delta^{13}C_{BF\text{-}CO}$ and $\Delta\delta^{13}C_{PF\text{-}}$ co                 | Amplified variability in $\Delta \delta^{18} C_{BF\text{-}CO}$                          | Intermediate-depth ventilation and mixing beneath a still-stratified surface layer.                                                           |
| Approaching
MIS M2
(~3.31 Ma)      | Increase in $\Delta\delta^{13}C_{BF-CO}$ and $\Delta\delta^{13}C_{PF-CO}$                                         | Decrease in $\Delta \delta^{18}C_{BF-CO}$                                               | Long-term warming and re-
establishment of a stratified ocean
with reduced vertical exchange.                                           |
| MIS M2
Glacial
(~3.30–3.28 Ma)     | $\delta^{13}C_{BF}$ and $\delta^{13}C_{CO}$
minima; followed by
recovery (stronger in $\delta^{13}C_{BF}$ ) | Peaks in $\Delta \delta^{18}O_{BF-CO}$ and $\Delta \delta^{18}O_{BF-PF}$ (deep cooling) | Onset: High-latitude cooling,
suppressed Atlantic Meridional
Overturning Circulation, intensified
stratification.                    |
|                                          |                                                                                                                   |                                                                                         | Termination: Increased deep ocean ventilation, potentially lagging surface reorganisation.                                                    |
| mPWP
Peak Warmth
(~3.264–3.025 Ma) | Stable but persistent vertical $\delta^{13}$ C gradients; high surface productivity but inefficient export.       | Generally negative $\delta^{18}O$ values (warming); muted vertical gradients.           | Strong thermal stratification, reduced overturning, and weakened thermocline ventilation limiting nutrient supply and carbon export.          |
| MIS KM2 Event
(within mPWP)           | Sharp collapse in all vertical $\Delta\delta^{13}C$ gradients.                                                    | Decline in all vertical $\Delta \delta^{18}$ O gradients (subsurface warming)           | Pulse of enhanced ventilation;
breakdown of vertical stratification,
possibly linked to high latitude
forcing and lateral advection. |
| Post-KM2 mPWP                            | Amplified variability in $\Delta \delta^{13} C_{BF-CO}$ and $\Delta \delta^{13} C_{BF-PF}$ .                      | Pronounced variability in $\Delta \delta^{18} O_{BF\text{-}CO}$                         | Dynamic shifts in nutricline depth
and reinvigorated biological pump;
recurrent deep-water mass
reorganisations.                     |

**Minor Comments**

• Ensure consistent use of "coccolith fraction" vs. "bulk fine fraction (<20 µm)".

We have now implemented this change. The term is defined once at its first mention as "fine fraction bulk carbonate ( $<20 \mu m$ , herein referred to as the coccolith fraction)" and is now used consistently as "coccolith fraction" throughout the remainder of the manuscript.

• Improve legibility of figure axes and legends (Figs. 3–5; Supplementary Figs. S1–S2) for print and grayscale viewing (i.e., increase font size slightly)

We have now increased the font sizes and improved the contrast for all axes and legends.

• A few recent studies on Indo-Pacific upwelling and Plio-Pleistocene productivity (e.g., Ford et al., 2022) could be cited to further contextualize results.

We have now incorporated the recommended reference (Ford et al., 2022; Ford et al., 2025) into Section 3.5 to better contextualize our findings on carbon cycling within the broader framework of Plio-Pleistocene ocean ventilation.

• Language is generally excellent; only minor copy-editing is needed to shorten some long sentences.

We thank the reviewer for their positive feedback. We have performed a thorough copy-edit of the manuscript to shorten long sentences and improve overall readability.

**Recommendation: Minor revisions**

This manuscript is a substantial and timely contribution to understanding tropical controls on Pliocene carbon cycling and orbital-scale climate feedbacks. Addressing the points above (particularly clarifying age-model uncertainty, refining coccolith isotope interpretation, and contextualizing productivity—export relationships) will further strengthen an already strong study.

We thank the reviewer for their positive and constructive feedback. We have carefully addressed all comments, with particular focus on clarifying the age-model uncertainty, refining the interpretation of the coccolith isotope records, and contextualizing the productivity-export relationships. We believe these revisions have strengthened the manuscript as suggested.

**References**

- Acker, J.G. and Leptoukh, G.: Online Analysis Enhances Use of NASA Earth Science Data. Eos, Transactions American Geophysical Union, 88, 14-17, 2007.
- Beal, L. M., Ruijter, W. P. M. D., Biastoch, A., Zahn, R., and 136, S. W. I. W. G.: On the role of the Agulhas system in ocean circulation and climate, Nature, 472, 429-436, doi:10.1038/nature09983, 2011.
- Bolton, C. T., Stoll, H. M., and Mendez-Vicente, A.: Vital effects in coccolith calcite: Cenozoic climate-pCO2drove the diversity of carbon acquisition strategies in coccolithophores?, Paleoceanography, 27, 10.1029/2012pa002339, 2012.
- de la Vega, E., Chalk, T. B., Wilson, P. A., Bysani, R. P., and Foster, G. L.: Atmospheric CO2 during the Mid-Piacenzian Warm Period and the M2 glaciation, Sci Rep, 10, 11002, 10.1038/s41598-020-67154-8, 2020.
- Ford, H. L., Burls, N. J., Jacobs, P., Jahn, A., Caballero-Gill, R. P., Hodell, D. A., and Fedorov, A. V.: Sustained mid-Pliocene warmth led to deep water formation in the North Pacific, Nature Geoscience, 15, 658-663, 10.1038/s41561-022-00978-3, 2022.
- Ford, H.L., Wrye, N., Wofford, A., Burls, N., Bhattacharya, T., Fedorov, A., Lakhani, K., Lyle, M., Lynch-Stieglitz, J. and Ravelo, A.C.: Warm equatorial upper ocean thermal structure during the mid-Pliocene warm period: A data-model comparison. Geophysical Research Letters, 52, 16, p.e2025GL114749, 10.1029/2025GL114749, 2025.
- Hermoso, M., Chan, I. Z. X., McClelland, H. L. O., Heureux, A. M. C., and Rickaby, R. E. M.: Vanishing coccolith vital effects with alleviated carbon limitation, Biogeosciences, 13, 301-312, 10.5194/bg-13-301-2016, 2016.
- Lisiecki, L. E. and Raymo, M. E.: A Pliocene-Pleistocene stack of 57 globally distributed benthic δ18O records, Paleoceanography, 20, 10.1029/2004pa001071, 2005.
- Meyers, S.: Astrochron: An R package for astrochronology, 2014.
- Westall, F. and Fenner, J.: Pliocene-Holocene polar front zone in the South Atlantic changes in its position and sediment-accumulation rates from holes 699A, 710C, and 704B. In Proc ODP, Sci Results, 114, pp. 609-646, 1991.
- Ziveri, P., Stoll, H., Probert, I., Klaas, C., Geisen, M., Ganssen, G. and Young, J., 2003: Stable isotope 'vital effects' in coccolith calcite. Earth and Planetary Science Letters, 210, 1-2, pp.137-149, 10.1016/S0012-821X(03)00101-8, 2003.